# Spatial variations and predictors of overweight/obesity among under-five children in Ethiopia: A geographically weighted regression analysis of the 2019 Ethiopian Mini Demographic and Health Survey

**Agmasie Damtew Walle** [1]*, **Shimels Derso Kebede** [2], **Jibril Bashir Adem** [3], **Ermias Bekele Enyew** [2], **Habtamu Alganeh Guadie** [4], **Teshome Bekana** [5], **Habtamu Setegn Ngusie** [6], **Sisay Maru Wubante** [7], **Sisay Yitayih Kassie** [8], **Addisalem Workie Demsash** [1], **Wabi Temesgen Atinafu** [9], **Tigist Andargie Ferede** [10]

**1** Department of Health Informatics, School of public health, Asrat Woldeyes Health Sciences Campus, Debre Berhan University, Debre Birhan, Ethiopia, **2** Department of Health Informatics, School of Public Health, College of Medicine and Health Sciences, Wollo University, Dessie, Ethiopia, **3** Department of Public Health, College of Medicine and Health Sciences, Arsi University, Asella, Ethiopia, **4** Department of Health System Management and, Health Economics, School of Public Health, College of Medicine and Health Science, Bahir Dar University, Bahir Dar, Ethiopia, **5** Department of Medical Laboratory Sciences, College of Health Sciences, Mattu University, Mettu, Ethiopia, **6** Department of Health Informatics, School of Public Health, College of Medicine and Health Sciences, Woldia University, Woldia, Ethiopia, **7** Department of Health Informatics, College of Medicine and Health Science, University of Gondar, Gondar, Ethiopia, **8** Department of Health Informatics, College of Medicine and Health Science, Hawassa University, Hawassa, Ethiopia, **9** Department of Public Health, College of Medicine and Health Sciences, Ambo University, Ambo, Ethiopia, **10** Department of Epidemiology and Biostatistics, College of Medicine and Health Science, University of Gondar, Gondar, Ethiopia

* agmasie89@gmail.com

## Abstract

### Background

Overweight/ obesity among under-five children is an emerging public health issue of the twenty-first century. Due to the quick nutritional and epidemiological change, non-communicable diseases, premature death, disability, and reproductive disorders have grown in low-income countries. Besides, little attention has been given. Therefore, we aimed to explore spatial variations and predictors of overweight/obesity among under-five children in Ethiopia using a geospatial technique.

### Methods

A total weighted sample of 3,609 under-five children was included in the study. A cross-sectional study was conducted using a nationally representative sample of the 2019 Ethiopia Mini Demographic and Health Survey data set. ArcGIS version 10.8 was used to explore the spatial variation of obesity. SaTScan version 9.6 software was used to analyze the spatial

**Data Availability Statement:** All relevant data are within the manuscript and its Supporting Information files.

**Funding:** The author(s) received no specific funding for this work.

**Competing interests:** The authors have declared that no competing interests exist.

**Abbreviations:** CI, Confidence Interval; EMDHS, Ethiopia Mini Demographic and Health Survey; GWR, Geographically Weighted Regression; LLR, Log-Likelihood Ratio; OLS, Ordinal Least Squares; SNNPR, Southern Nations, Nationalities, and Peoples' Region; WHO, World Health Organization.

cluster detection of overweight/obesity. Ordinary least square and geographically weighted regression analysis were employed to assess the association between an outcome variable and explanatory variables. A p-value of less than 0.05 was used to declare it statistically significant.

## Results

The spatial distribution of overweight/obesity among under-five children in Ethiopia was clustered (Global Moran's I = 0.27, p-value<0.001). The significant hot spot areas or higher rates of childhood obesity, were found in Southern Amhara, Northwest Somalia, Border of Harari, central Addis Ababa, Eastern SNNPR, and Northwestern Oromia region. In spatial SaT Scan analysis, 79 significant clusters of overweight/obesity were detected. The primary clusters were located in SNNPR, Oromia, and Addis Ababa (RR = 1.48, LLR = 31.40, P-value < 0.001). In the geographically weighted regression analysis, urban residence, cesarean section, rich households, and female children were statistically significant predictors.

## Conclusions

Overweight or obesity among under-five children show spatial variations across Ethiopian regions. GWR analysis identifies cesarean section, wealth index, urban residence, and child sex as significant predictors. The Ministry of Health and Ethiopian Public Health Institute should target regions with these contributing predictors, promoting localized physical education, health education campaigns, and ongoing community monitoring to encourage active lifestyles and reduce sedentary behaviors among children.

## Introduction

Overweight or obesity is defined by the World Health Organization (WHO) as an abnormal or excessive accumulation of fat that may be hazardous to one's health [1, 2]. The combined impacts of childhood obesity and overweight are one of the most important public health challenges of the twenty-first century [3]. Childhood obesity and overweight are becoming more widespread in all countries [4].

Globally, the prevalence of obesity among under five children was 7% in 2012, with projections indicating it will rise to nearly 11% by 2025 [5]. According to the 2018 Global Nutrition Report, the prevalence of overweight among under five children globally is 1.87%, affecting approximately 8.23 million children [6, 7]. The rate of overweight varies across regions, with Europe having the highest prevalence at 2.7%, followed by Africa at 2.3%, and the Americas at 0.8% [7].

The majority of overweight and obese children are found in developing countries, where the rate of increase is over 30% faster than in industrialized nations [8]. In Africa, the number of overweight and obese children has almost doubled, increasing from 5.4 million in 1990 to 10.3 million in 2014 [3, 9]. According to the Demographic and Health Surveys conducted in SSA, about 6.8% of children between the ages of 0 and 59 months were overweight or obese in 2010 [10]. These trends are reflective of rapid changes in nutrition and epidemiology, leading to a significant rise in non-communicable diseases linked to over nutrition [5]. Despite the plateauing of overweight and obesity rates among infants, children, and adolescents in high-

income countries, the prevalence remains higher in low- and middle-income countries, including many African settings [3, 11].

In Ethiopia, a low-income country in Sub-Saharan Africa, childhood obesity has not been given significant attention and is not yet considered an emergent public health concern [1, 12]. According to UNICEF's annual report, the prevalence of childhood obesity in Ethiopia increased from 1.7% to 3.6% in 2017 [13]. Specifically, studies have shown that the prevalence of childhood obesity and overweight was 10.7% in Hawassa City, with 3.4% of children being overweight and 7.3% being obese [14]. In Gondar Town, a study revealed that the combined prevalence of overweight and obesity among children and adolescents was 13.8% and 11.3%, respectively [15]. These figures highlight the growing burden of childhood obesity in Ethiopia, necessitating immediate public health interventions.

Although childhood obesity is recognized as a public health issue in Ethiopia, the country's health efforts remain focused on addressing undernutrition and infectious diseases, with limited emphasis on tackling obesity [2]. Limited attention from health policies, cultural perceptions of body weight, resource constraints, and lack of comprehensive data contribute to this oversight [16, 17]. To address the growing burden of childhood obesity, Ethiopia needs to expand its public health focus to include obesity prevention and intervention programs, learning from other low- and middle-income countries that have successfully integrated efforts to combat both undernutrition and obesity.

Childhood obesity and/or overweight negatively influence physical and mental health, with an increased chance of remaining overweight into adulthood. Obesity may raise the risk of developing non-communicable diseases (NCDs), such as diabetes and cardiovascular disease, during this period [18, 19]. Furthermore, overweight and obese children face an increased risk of developing non-communicable diseases. Childhood obesity and overweight are associated with a higher mortality rate compared to underweight, along with breathing difficulties, an elevated risk of fractures, hypertension, early indicators of cardiovascular disease, insulin resistance, and psychological impacts [6].

Previous research found that numerous factors could be linked to childhood overweight/obesity in children under the age of five. These include socioeconomic status, the child's gender, birth weight, and birth rank, residence, the child's age, maternal education level, time spent watching television >2 hours per day, marital status, smoking during pregnancy, body mass index (BMI) of parents, high dietary diversity, and sweet food consumption [1, 3, 6, 7, 19].

Under-five obesity has become a double burden in Ethiopia, existing alongside undernutrition and emerging as a significant public health issue [3]. Previous research in Ethiopia has not applied spatial regression analysis, which is particularly important due to the country's diverse geography, socioeconomic disparities, and unequal access to healthcare and food markets. The varied lifestyles, diets, and healthcare access across Ethiopia's highlands, lowlands, urban, and rural areas influence childhood obesity differently, making spatial analysis highly relevant.

Spatial analysis, particularly using Geographically Weighted Regression (GWR), is crucial for understanding the variations in childhood overweight and obesity across Ethiopia. This method helps identify obesity hotspots and areas with limited healthcare access, enabling policymakers to design targeted, region-specific interventions. By examining factors such as socioeconomic status, dietary patterns, and physical activity, GWR provides a detailed understanding of how the individual as well as the community related factors vary across different regions, including highlands, lowlands, urban centers, and rural areas. This approach allows for more effective resource allocation and intervention strategies tailored to the unique needs of each region.

Therefore this study aimed to assess the spatial variations and predictors of overweight/obesity among under five children by using GWR analysis of 2019 Demographic and Health Survey (DHS) data set. The findings will help policymakers and health planners design targeted interventions, allocate resources more efficiently, and address regional disparities in childhood obesity. By providing a nuanced understanding of regional patterns and underlying mechanisms, this study aims to improve public health outcomes for under five children in Ethiopia and contribute to long-term efforts in managing and reducing childhood obesity, while addressing the growing burden of non-communicable diseases.

## Methods and materials

### Study design, period, and area

A community-based cross-sectional study was conducted by the Central Statistical Agency (CSA) from March 21, 2019, to June 28, 2019., in Ethiopia using the EMDHS 2019 data set [20]. Ethiopia is a country located in the Horn of Africa, known for its rich history, diverse cultures, and ancient civilizations. It has a federal system of government and is divided into nine regional states: Afar, Amhara, Benishangul-Gumuz, Gambella, Harari, Oromia, Somali, Southern Nations, Nationalities, and Peoples' Region (SNNPR), and Tigray. Additionally, there are two chartered cities: Addis Ababa, the capital city and the largest in Ethiopia, and Dire Dawa [21].

### Source and study population

The source population was all under five year's old children within five years before the survey in Ethiopia. While the study population specifically comprised living children aged 0 to 59 months, as recorded in the selected Enumeration Areas (EAs) during the survey period. Data for this study were derived from the children's dataset (KR) of the 2019 Ethiopian Mini Demographic and Health Survey (EMDHS). The study included a weighted sample of 3,609 children under five. Children with missing weight-for-height z-score information and observations from enumeration regions with zero coordinates were excluded from the analysis.

### Data collection tool and sampling procedures

This study was used the 2019 Ethiopian Mini Demographic and Health Survey (EMDHS) data set. The sample for the 2019 EMDHS was stratified and selected in two steps. In the first stage, a total of 305 EAs (93 in urban areas and 212 in rural areas) were selected with probability proportional to EA size (based on the 2019 PHC frame) and with independent selection in each sampling stratum [22]. In the second stage, lists of households served as a sampling frame for the selection of households. We retrieved the data from the DHS website (www.dhsprogram.com) after we allowed it by the measuring program. The detailed sampling procedure has been presented in the 2019 EMDHS report [20].

### Study variables

Overweight/obesity in children under the age of five was outcome variable for this study, and it was scored as "1," else marked as "0". According to the World Health Organization (WHO), child growth monitoring charts, if the kid's weight-for-height z-score is greater than plus 2 (+2.0) standard deviations (SD) over the mean, overweight and/or obesity is proclaimed [19].

Child age, child sex, birth order, twin child, delivered by cesarean section, mother's education status, current age, marital status, child ever born, age of mother at first birth, number of

household members, sex of head of households, wealth index, residence, region, and cluster altitude, and contraceptive use were the independent variables.

## Data management and statistical analysis

The descriptive analysis was carried out using the statistical program STATA version 14. Arc-GIS 10.8 and SaTScan 9.6 were used for the spatial analysis. STATA was used to tabulate the weighted proportions of outcome variables and prospective predictor variables, which were then exported to Excel and imported into ArcGIS 10.8 for additional analysis. Imputation techniques were employed to estimate or replace missing values based on existing data and this imputation methods helps to maintain the integrity and completeness of the dataset, allowing for more robust and reliable analysis.

## Spatial analysis

At the national level, the spatial autocorrelation (Global Moran's I) statistics were utilized to determine if the overweight/obesity distribution is random or not. Moran's, I value close to -1 indicates that overweight/obesity is dispersed, whereas Moran's I close to +1 indicates over-weight/obesity is clustered, and Moran's I close to 0 revealed that overweight/obesity is randomly distributed. A statistically significant Moran's I ($p < 0.05$) result demonstrated that overweight/obesity is not random of the null hypothesis [23].

Hot spot analysis was computed to measure how spatial autocorrelation varies over the study location by calculating Getis-ordGi* statistic for each area. To determine whether there was substantial clustering, the Z-score and p-values were calculated. Statistical values with high Getis-ordGi* indicate a "hotspot" whereas low Getis-ordGi* means a "cold spot" [24].

In unsampled areas, empirical Bayesian kriging interpolation was employed to predict over-weight/obesity [25]. Spatial SaTScan analysis was used to perform cluster analysis to detect the more likely clusters by computing the relative risk (RR) and testing the statistical significance. The outcome variable has a Bernoulli distribution, so the Bernoulli model was used by applying Kuldorff's method for purely spatial analysis. The scanning window that moves across the study area in which children give overweight/obese was taken as a case and those children who give normal body weight were taken as a control to fit the Bernoulli model. The default maximum spatial cluster size of < 50% of the population was used as an upper limit and most likely clusters was identified by using p-values and likelihood ratio tests based on 999 Monte Carlo replications [22, 24].

## Spatial regression

To identify the predictors of spatial heterogeneity in overweight/obesity, spatial regression modeling was conducted. The Ordinary Least Squares (OLS) model, a global statistical method, was used to evaluate and explain the relationship between the outcome variable (obesity) and the explanatory variables [26]. OLS also served as a diagnostic tool to determine which predictors should be included in the Geographic Weighted Regression (GWR) model [27].

Before proceeding with the GWR analysis, several assumptions were tested, including non-stationarity, residual spatial autocorrelation, model bias, and multicollinearity. These were assessed using the Koenker test, Moran's I, Jarque-Bera statistics, and the Variance Inflation Factor (VIF), respectively. The GWR is a local spatial statistical approach that accounts for non-stationarity in the relationship between the outcome and predictors across Enumeration Areas (EAs) [27]. A statistically significant Koenker test result supported the use of GWR analysis. In the GWR model, the coefficients of the predictors vary across the study area, allowing for more localized insights. Mapping these GWR coefficients for the predictors helps inform

targeted interventions. The model with the lowest corrected Akaike Information Criterion (AIC) and the highest adjusted $R^2$ was selected as the best fit for the data.

### Ethics approval and consent to participate

The study did not require ethical review or participant agreement letter because it was a secondary data analysis of publicly accessible survey data from the MEASURE DHS program. We requested DHS Program for permission to obtain and use the data for this study from their website http://www.dhsprogram.com. Procedures for DHS public-use datasets that have been approved by the Institutional Review Board do not in any manner make it possible to identify respondents, households, or sample communities.

The data file does not contain any personal information like names or home addresses. The data file contains a PSU number for each enumeration area (Primary Sampling Unit), but there are no labels to denote the names or locations of the PSUs. In surveys that gather GIS coordinates in the field, the measured coordinates are randomly displaced within a wide geographic area so that specific enumeration areas cannot be recognized. The measured coordinates are only for the enumeration area (EA) as a whole, not for individual households.

## Results

### Characteristics of the respondents and study children

A weighted total of 3,609 children were included in the study. Out of the 3,609 children, 1,129 (3.13%) were overweight/obese. 1,875 (51.97%) of children were males. Of the total participants, 2,626 (37.99%) mothers were from the Oromia region and 2,694 (68.09%) were from the rural areas. Of the participants, 1,549 (43.00%) children were the age of above 25 months, and 3,101 (44.87%) of the mothers did attain primary education. About 3,155 (45.64%) of the mothers were the richest household index quintiles, Moreover, 3,400 (94.24%) of children were delivered by cesarean section (Table 1).

### Prevalence of overweight/obesity among under-five children in Ethiopia

The prevalence of overweight/obesity among under-five children was 3.13% (95%CI: 2.83%-4.92%). The highest prevalence of overweight/obesity among under-five children was observed in Addis Ababa (4.85%), Harari (3.95%), Oromia (3.78%), and SNNPR (3.05%) regions. Whereas, the lowest prevalence of overweight/obesity was observed in Somali (1.36%), Gambela (1.68%), and Afar (2.13%) regions (Fig 1).

### Spatial analysis

**Spatial autocorrelation.** The global spatial autocorrelation (Global Moran's I) showed that there were clustered patterns of overweight/obesity among under-five children across Ethiopia (Global Moran's I = 0.27, p-value<0.001). Moreover, Z-score is also positive (5.94), indicating that the observed clustering is less than 1% likely to be the result of random chance (Fig 2).

### Hot spot and cold spot analysis

The Getis Ord-Gi* statistics identified the significant hot spots (higher risk) and cold spots areas of overweight/obesity among under-five children. The red colors indicated the significant hotspot areas of overweight/obesity among under five children, this showed that areas with higher rates of children obesity, which were found in the Southern part of Amhara, Northwest Somalia, Border of Harari, central Addis Ababa, Eastern part of SNNPR and Northwestern part of Oromia region of Ethiopia (Fig 3).

**Table 1. Socio-demographic characteristics of study participants in Ethiopia, EMDHS 2019.**

| Variables | Weighted frequency | Percentage |
|---|---|---|
| **Region** | | |
| Tigray | 490.33 | 7.09 |
| Afar | 68.81 | 1.00 |
| Amhara | 1,515.53 | 21.92 |
| Oromia | 2,625.92 | 37.99 |
| Somali | 355.48 | 5.14 |
| Benishangul | 76.24 | 1.10 |
| Snnpr | 1,313.15 | 19.00 |
| Gambela | 30.17 | 0.44 |
| Harari | 20.89 | 0.30 |
| Addis Adaba | 366.46 | 5.30 |
| Dire Dawa | 49.85 | 0.72 |
| **Residence** | | |
| Urban | 2,205.85 | 31.91 |
| Rural | 4,707.00 | 68.09 |
| **Educational level** | | |
| No education | 2,350.78 | 34.01 |
| Primary | 3,101.58 | 44.87 |
| Secondary | 1,006.05 | 14.55 |
| Higher | 454.44 | 6.57 |
| **Marital status** | | |
| Single | 2,268 | 32.61 |
| Married | 4,291 | 61.71 |
| Widowed | 52 | 0.75 |
| Divorced | 343 | 4.93 |
| **Wealth status** | | |
| Poor | 2,475.85 | 35.82 |
| Rich | 3,154.77 | 45.64 |
| Middle | 1,282.22 | 18.55 |
| **Household head** | | |
| Male | 5,591.35 | 80.88 |
| Female | 1,321.49 | 19.12 |
| **Twin birth** | | |
| Single birth | 3,817.92 | 98.32 |
| 2nd of multiple | 63.02 | 1.62 |
| 3rd of multiple | 2.21 | 0.06 |
| **Sex of child** | | |
| Male | 1,875.39 | 51.97 |
| Female | 1,733.18 | 48.03 |
| **Contraceptive use** | | |
| Using modern method | 1,929.53 | 27.91 |
| Using traditional method | 45.58 | 0.66 |
| Does not use | 4,937.74 | 71.43 |
| **Child delivered by cesarean section** | | |
| No | 3,400.83 | 94.24 |
| Yes | 207.74 | 5.76 |
| **Age of child in a month** | | |

*(Continued)*

**Table 1.** (Continued)

| Variables | Weighted frequency | Percentage |
|-----------|-------------------|------------|
| Less than 6 | 579.93 | 16.01 |
| Between_6–24 | 1,479.40 | 40.99 |
| Greater than_25 | 1,549.24 | 43.00 |
| **Birth interval** | | |
| Less than_23 | 631 | 20.57 |
| Greater than_23 | 2,436 | 79.43 |

**Spatial interpolation.** The ordinary kriging spatial interpolation method revealed that predicting overweight/obesity in unobserved areas and the predicted overweight/obesity increases from green to red-colored areas. High-risk areas of predicted childhood overweight or obesity are indicated by the red color, and low-risk areas are indicated by the green color. Accordingly, central, Southwestern, and Southeastern parts of SNNPR, Oromia, Addis Ababa, and the Southwestern part of Harrari were predicted as the riskiest areas for overweight/obesity among under-five children compared with other regions. Whereas the predicted low-risk

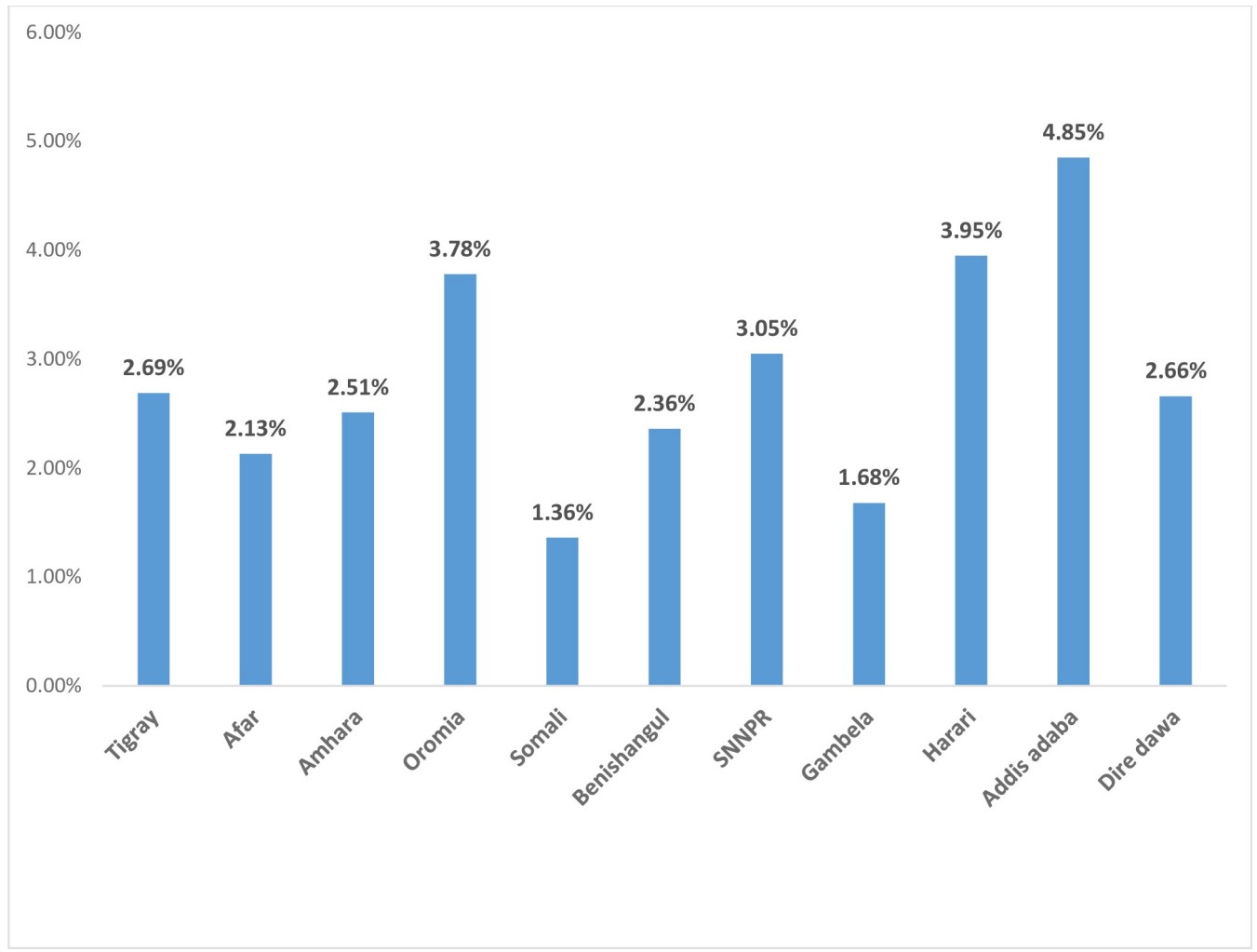

**Fig 1. The prevalence of overweight/obesity among under-five children across regions in Ethiopia, EMDHS 2019.**

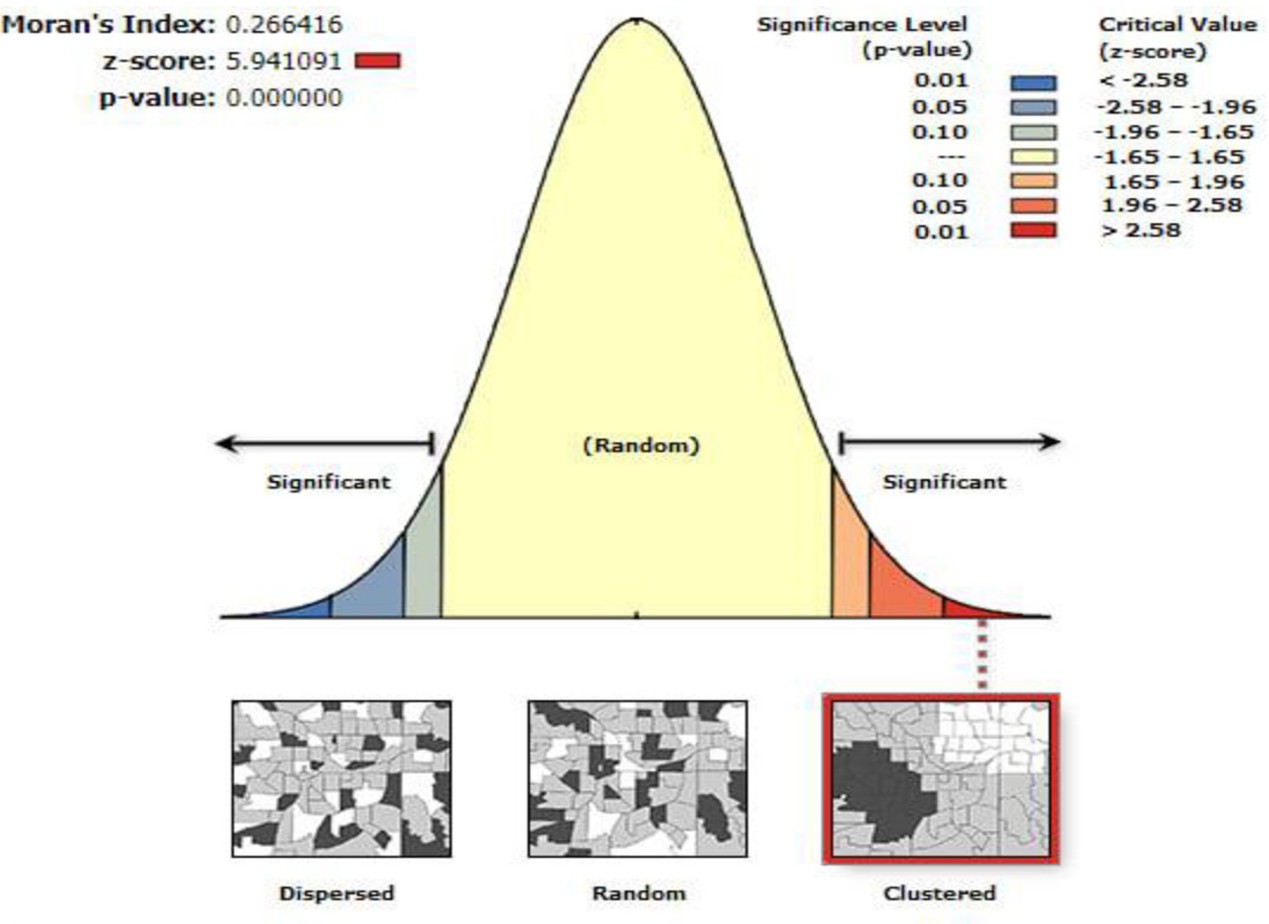

**Fig 2. Spatial autocorrelation of overweight/obesity among under-five children in Ethiopia, EMDHS 2019.**

areas for overweight/obesity were identified in Somali, Afar, the Border of Beneshangul Gumuz, and Gambella region (Fig 4).

## Spatial SaTScan analysis

The Spatial SaTScan analysis revealed that, a total of 79 significant clusters of overweight/obesity were identified. The most likely (primary) clusters were located in SNNPR, Oromia, and Addis Ababa at (7.000000 N, 39.000000 E) geographical location with a 312.78 km radius. Children aged 0–5 years who lived in the primary cluster were 1.48 times more likely than those who lived outside the window to experience overweight/obesity (RR = 1.48, LLR = 31.40, P-value < 0.001) (Fig 5) (Table 2).

## Factor affecting the spatial variation of overweight/obesity among under-five children

**Ordinary least square regression.** After the candidate explanatory factors were fitted with the ordinal least squares (OLS) model, the model was able to explain 56.7% (Adjusted R

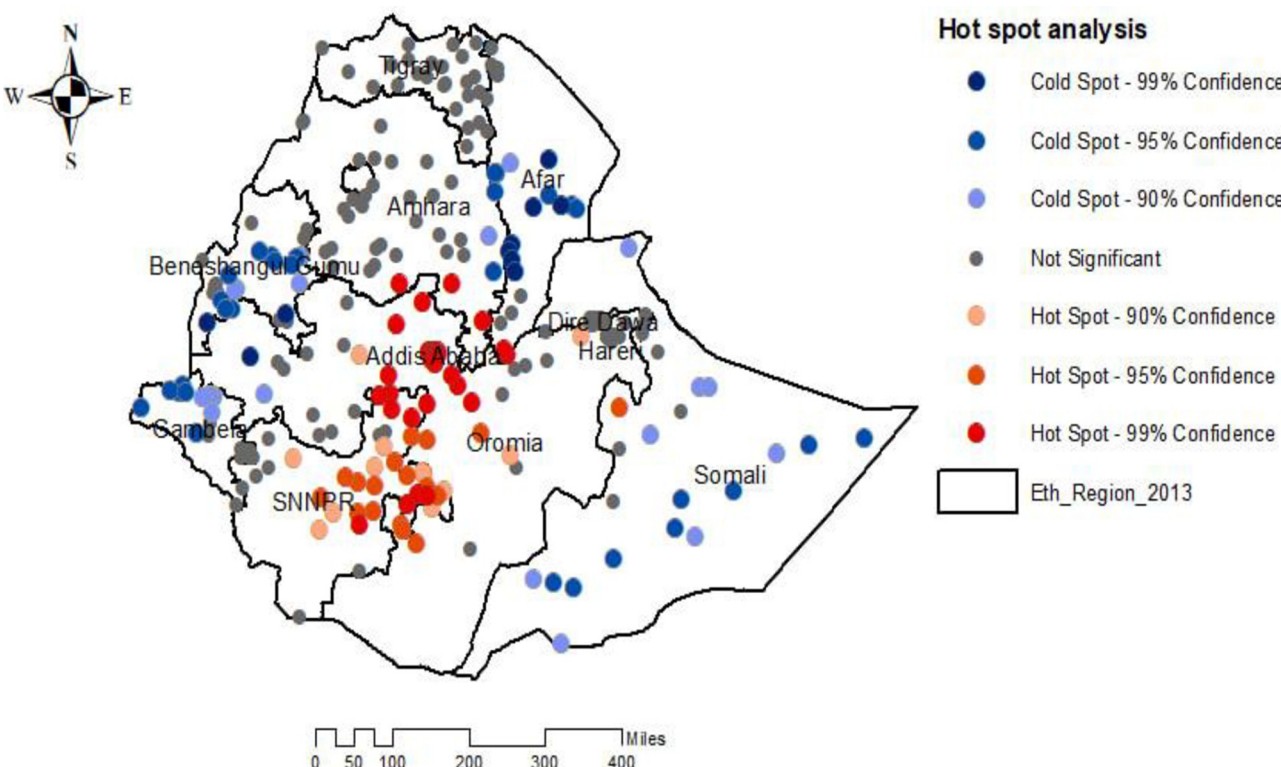

**Fig 3. Hot spot analysis of overweight/obesity among under-five children in Ethiopia, EMDHS 2019.**

square = 0.567) of the variation in overweight/obesity. Moreover, all of the OLS method's assumptions were met. All of the coefficients were found to be statistically significant when the robust probability was applied to assess their significance. The Joint Wald statistic was also statistically significant ($p < 0.01$), indicating that the total model was significant. Since, the variance inflation factor (VIF) $< 7.5$, multi-collinearity between explanatory variables did not exist. In addition, the Jarque-Bera statistic was non-significant indicating the model residuals were normally distributed.

According to the statistical significance of the Koenker statistics ($p < 0.01$), there was a non-stationary or heterogeneous relationship between the independent and dependent variables across the research areas. Given that the relationship between the independent and dependent variables is assumed to be spatially heterogeneous throughout the region (as demonstrated by the non-stationarity of the relationship in the Koenker statistics), it was suggested that GWR be utilized. The proportion of children who lived in urban, the proportion of children who were delivered by cesarean section, the proportion of children who were from rich households, and the proportion of female children were significantly associated with the percentage of overweight/obesity among under-five children in the OLS model (Table 3).

**Geographically weighted regression.** To strengthen the model in case of non-stationarity between predictors and overweight/obesity, this study conducted GWR, because the OLS regression only identified the predictors of overweight/obesity and it is a global model that assumes the relationship between each explanatory variable and the outcome variable (obesity) is stationary across the study area. Selected predictor variables were fitted to the spatially weighted regression model. The data were also fitted for model compression using both the Geographical Weighted Regression (GWR) and Ordinary Least Square (OLS) approaches (Table 4).

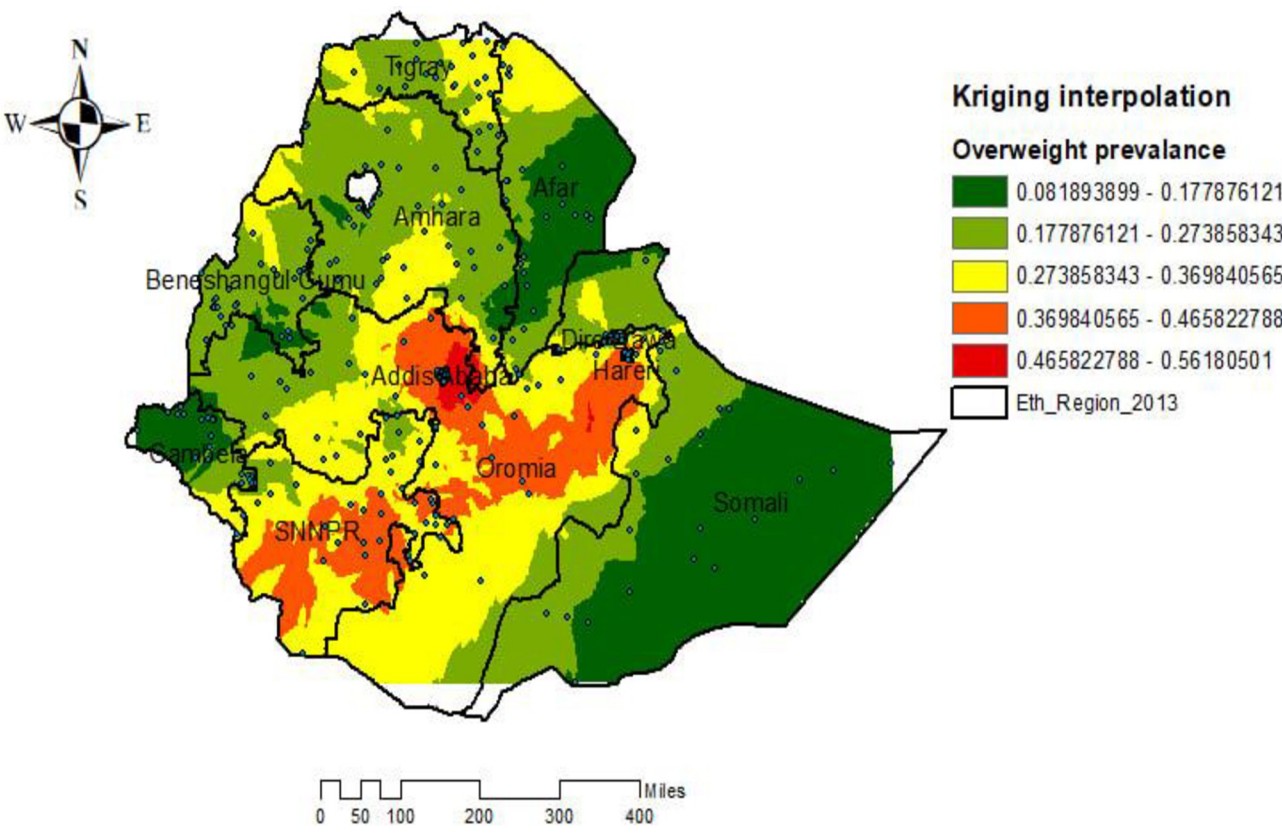

**Fig 4. Spatial interpolation of overweight/obesity among under-five children in Ethiopia, EMDHS 2019.**

Model comparison was measured using the corrected Akakie Information Criteria (AICc) and adjusted R2. As shown in Table 5, geographically weighted regression was the best-fit model with AICc of -234.45 compared with -223.54 in OLS model, the GWR model is best explained by the predictor variables for overweight/obesity among under-five age group children with an adjusted R2 value of 59.2% compared to OLS adjusted R2 value of 56.7%.

The proportion of women living in urban areas, the percentage of children delivered via cesarean section, the percentage of women from affluent households, and the percentage of male children were all regarded as explanatory variables in the GWR model and had a good R2 in the exploratory analysis of the geographically weighted regression analysis. There was a positive correlation between the percentage of women living in urban areas and the percentage of under-five children who were overweight or obese. As the proportion of women who had lived in urban was increased, the percentage of overweight/obesity among under-five children increased in the entire Amhara, Tigray, Afar, and Addis Ababa regions (Fig 6).

The proportion of children who were delivered by cesarean section was significantly associated with the increased risk of overweight/obesity among under-five children, with the highest effect on mothers delivered by cesarean section observed in northeast Oromia, Dire Dawa, South Afar, and Northwest regions (Fig 7).

The proportion of mothers in the richest household wealth status showed strong and positively associated with increased risk of overweight/obesity among under-five children in Beneshangul Gumuz, Gambela, SNNPR, the western part of Addis Ababa, and East Somali regions (Fig 8).

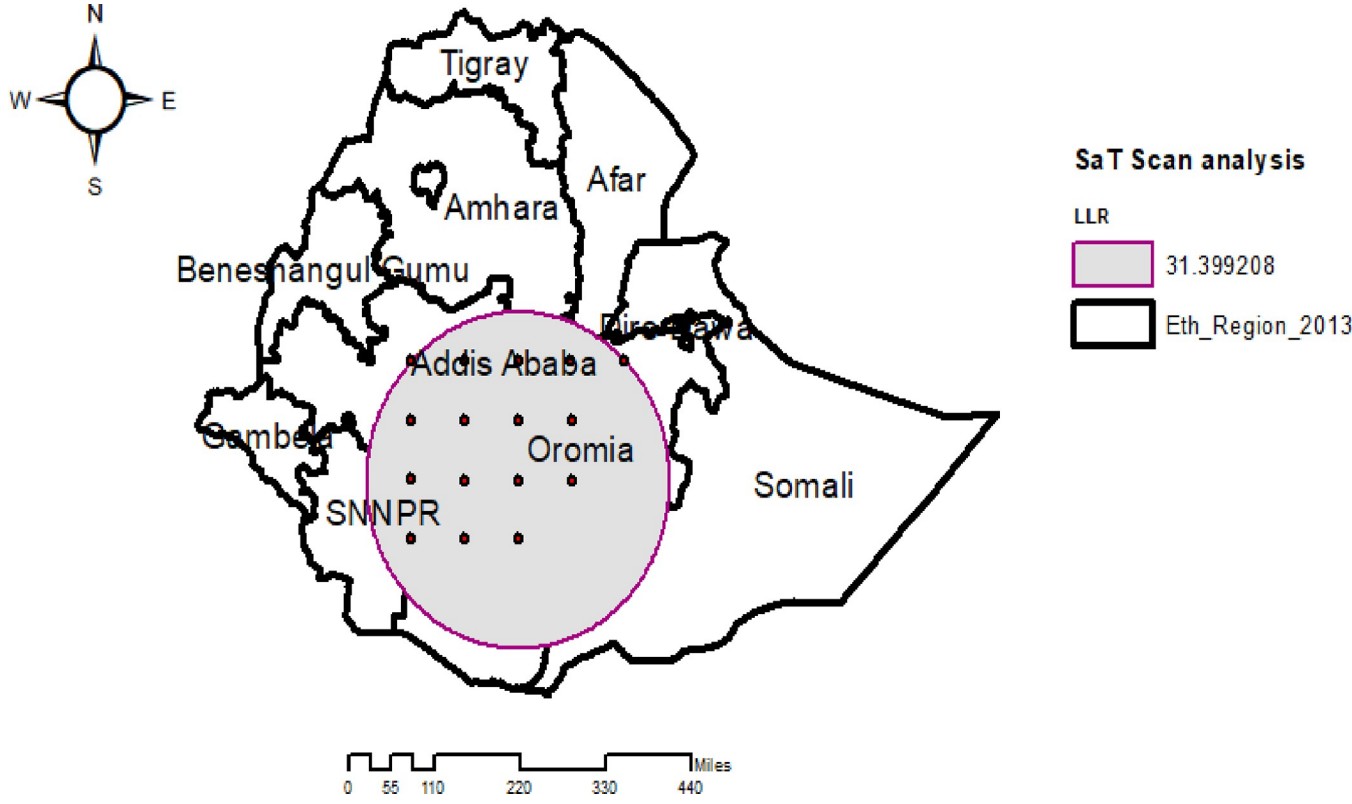

**Fig 5. Spatial Sat Scan analysis of overweight/obesity among under-five children in Ethiopia, EMDHS 2019.**

The proportion of female children had a significant negative association with overweight/obesity among under-five children in the entire Tigray, Amhara, Northwest Afar, and Bene-shangul Gumuz regions (Fig 9).

## Discussion

The purpose of this study was to examine the geographical variations in overweight and obesity, and predictors among Ethiopian under five children. Overweight and obesity in this age group remain significant public health concerns in Ethiopia. According to this study, the prevalence of overweight or obesity in Ethiopian children under five years old was 3.13% (95%CI: 2.84%-4.92%). This finding was lower than in a study conducted in Cameroon (8%) [28], Kenya (20.3%) [29], Sub-Saharan Africa (6.8%) [10], Nigeria (23.6%), Brazil (9.7%). This study's finding was also lower than the global prevalence of overweight/obesity (7%) [30].

The possible explanation for this discrepancy could be Ethiopia's lower socio-economic status limits access to processed foods and sedentary lifestyles, which are contributors to higher

**Table 2. Significant clusters of overweight/obesity among under-five children in Ethiopia, EMDHS 2019.**

| Cluster | The enumeration area (cluster) identified | Coordinate/radius | Population | Case | RR | LRR | P-Value |
|---------|-------------------------------------------|-------------------|------------|------|-----|------|---------|
| 1 | 117, 181, 183, 185, 186, 187, 178, 180, 182, 184, 189, 190, 110, 111, 116, 175, 90, 113, 114, 171, 174, 176, 177, 179, 203, 205, 102, 103, 104, 115, 172, 188, 197, 89, 191, 196, 204, 101, 256, 257, 258, 259, 260, 261, 262, 263, 264, 265, 266, 267, 268, 269, 270, 271, 272, 273, 274, 275, 276, 277, 278, 279, 280, 91, 95, 96, 173, 192, 198, 199,112, 105, 28, 41, 87, 98, 106, 127, 88 | (7.000000 N, 39.000000 E) / 312.78 km | 1688 | 640 | 1.48 | 31.40 | <0.001 |

**Table 3. Ordinary Least Squares (OLS) model summary and diagnostics of overweight/obesity among under-five children in Ethiopia, EMDHS 2019.**

| Variable | Coefficient | Robust standard error | Robust t statistics | Robust probability | VIF |
|---|---|---|---|---|---|
| Intercept | 0.31 | 0.05 | 6.64 | <0.01 | . . … |
| Urban resident | 0.07 | 0.03 | 2.36 | <0.01 | 2.36 |
| Sex of children (female) | -0.19 | 0.06 | 3.05 | <0.01 | 1.01 |
| Rich household | 0.22 | 0.04 | 5.82 | <0.01 | 3.16 |
| Child delivered by cesarean section | 0.01 | 0.05 | 2.33 | <0.05 | 1.80 |
| **Ordinary least squire regression diagnostics** | | | | | |
| Number of observations | 305 | | Akaike's Information Criterion (AICc) | | -223.54 |
| Multiple R-squared | 0.578 | | Adjusted R-Squared | | 0.567 |
| Joint F-Statistic | 16.22 | | Prob(> F), (4300) degrees of freedom | | <0.01 |
| Joint Wald Statistic | 70.91 | | Prob(>chi-squared), (4) degrees of freedom | | <0.01 |
| Koenker (BP) Statistic | 18.25 | | Prob(>chi-squared), (4) degrees of freedom | | <0.01 |
| Jarque-Bera Statistic | 3.86 | | Prob(>chi-squared), (2) degrees of freedom | | 0.15 |

overweight/obesity rates in more affluent regions. Cultural factors play a significant role, with traditional Ethiopian diets emphasizing whole grains, vegetables, and legumes that support healthier eating habits from an early age. These cultural feeding practices, along with preferences for local, less processed foods, likely contribute to lower rates of overweight/obesity among Ethiopian children. Methodological differences in how overweight/obesity is defined and assessed, as well as variations in data collection methods across studies, also influence reported prevalence rates. Understanding these factors not only highlights Ethiopia's distinct nutritional landscape but also informs targeted interventions that promote healthy lifestyles and prevent childhood overweight/obesity in the context of local cultural and socio-economic realities [1].

According to the GWR analysis, there is a significant correlation between the percentage of children delivered via cesarean section and the higher risk of overweight/obesity among children under five. The regions with the highest effect on mothers who deliver via cesarean section are northeast Oromia, Dire Dawa, South Afar, and Northwest. This finding was in line with a population based study done in China [31], Germany [32], and East China [33].

The possible reason could be infants born via cesarean section miss out on exposure to the maternal vaginal microbiota during birth. Vaginal bacteria, such as Bacteroides and bifidobacteria, are believed to contribute to the development of the infant's gut microbiome, potentially impacting metabolic health and obesity risk later in life. Studies have shown that children

**Table 4. Geographically Weight Regression (GWR) model summary and diagnostics of overweight/obesity among under-five children in Ethiopia, EMDHS 2019.**

| Explanatory variables | Urban residents, children who were delivered by cesarean section, rich households, and male children |
|---|---|
| Residual squire | 6.97 |
| Effective number | 31.42 |
| Sigma | 0.15 |
| AICc | -234.45 |
| Multiple R2 | 0.644 |
| Adjusted R2 | 0.592 |

N.B AICc: Akaike's Information Criterion

**Table 5. Model comparison of OLS and GWR model.**

| Model comparison | OLS model | GWR model |
|---|---|---|
| Akakie Information Criteria | -223.54 | -234.45 |
| Adjusted R2 | 56.7% | 59.2% |

delivered by cesarean section have lower levels of Bacteroides and bifidobacteria compared to those born vaginally. These bacteria are thought to offer protection against future obesity [34]. Moreover, women who have cesarean deliveries often have higher pre-pregnancy body mass indices (BMIs) or may develop gestational diabetes during pregnancy [33]. Cesarean births have also been associated with lower rates of early breastfeeding and reduced umbilical leptin concentrations, both of which have been linked to an increased risk of developing obesity later in life [1].

In the same manner, the proportion of mothers in the wealthiest household status were in Beneshangul Gumuz, Gambela, SNNPR, the western region of Addis Ababa, and East Somalia shown a strong and positive correlation with an elevated risk of overweight/obesity among under five children. Previous studies done in Ethiopia [14], sub-Saharan Africa [35], and East Africa [19] reported that children born from rich households were at high risk of being over-weight and/or obese. The possible reason could be wealthier families may have greater financial means to afford high-calorie, low-nutrient foods, which are more readily available in

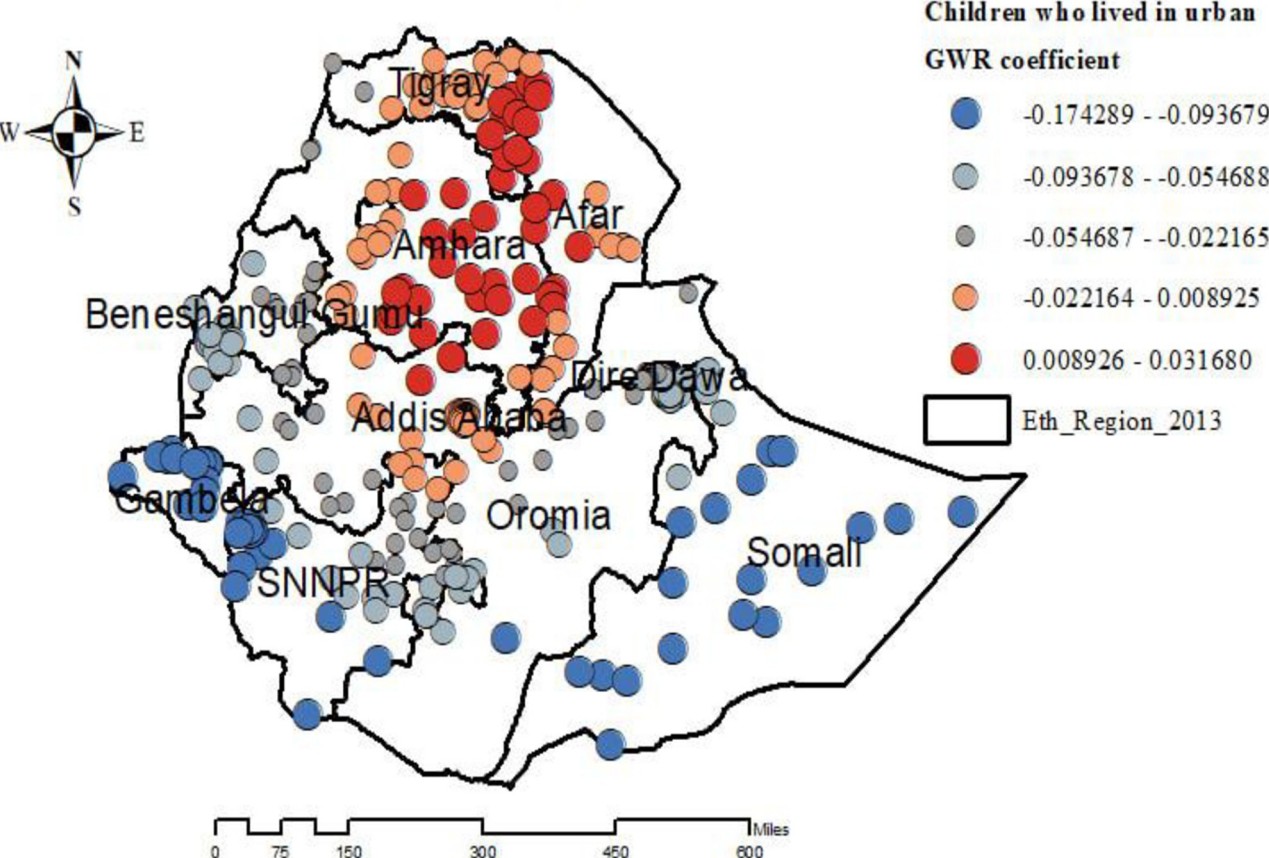

**Fig 6. Geographically weighted regression coefficients of urban residence to predict the hotspots of overweight/obesity among under-five children in Ethiopia.** EMDHS 2019.

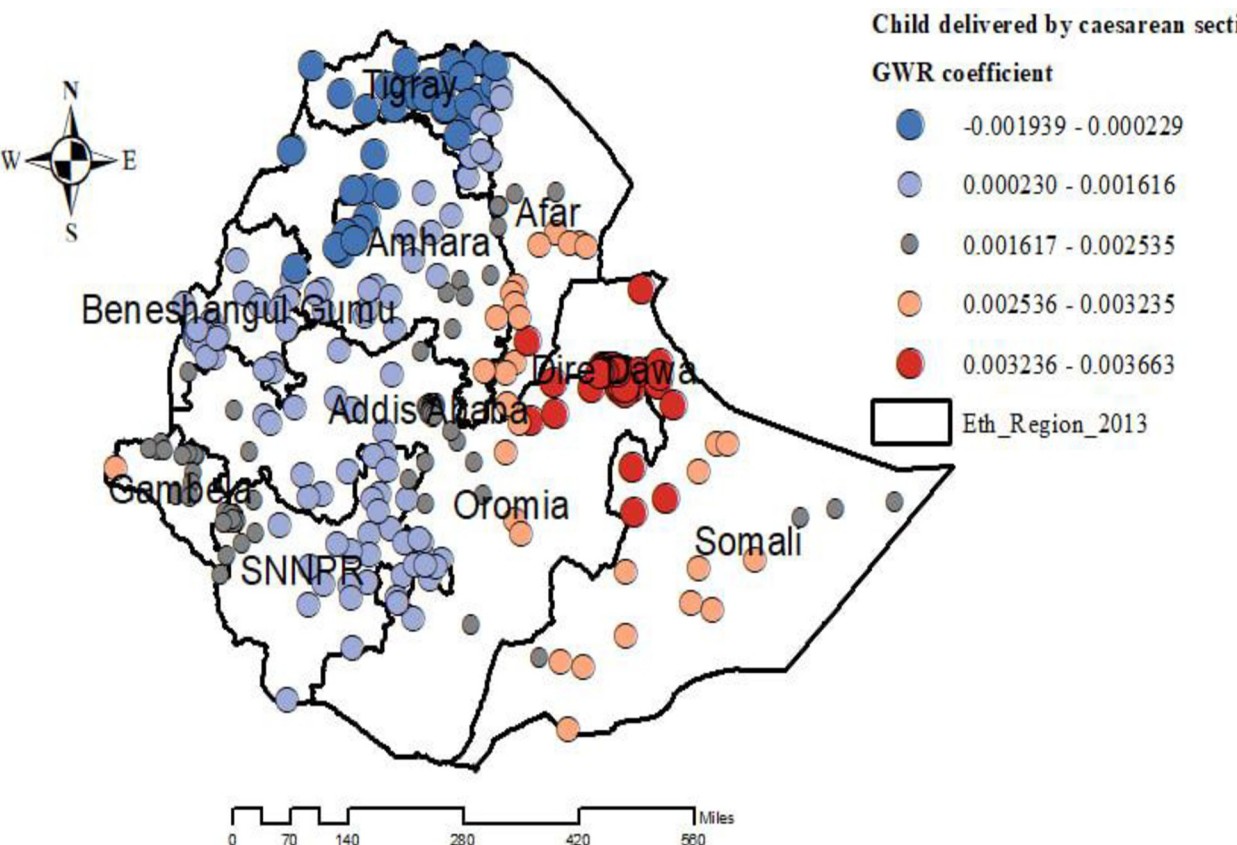

**Fig 7. Geographically weighted regression coefficients of child delivered by cesarean section to predict the hotspots of overweight/obesity among under-five children in Ethiopia.** EMDHS 2019.

urban and affluent areas. Additionally, children from these households may experience reduced physical activity due to changes in lifestyle, such as increased screen time and less outdoor play, further contributing to weight gain. This pattern has been observed in studies from Ethiopia, sub-Saharan Africa, and East Africa, where the nutrition transition shifting dietary patterns toward more processed foods plays a significant role in the rising prevalence of childhood overweight and obesity [36].

There was a positive correlation between the proportion of women living in urban areas and the proportion of under-five children who were overweight or obese. As the proportion of women who had lived in urban was increased, the percentage of overweight/obesity among under-five children increased in the entire Amhara, Tigray, Afar, and Addis Ababa regions. This finding was in line with a previous study in Ethiopia [3], Poland [37], and Peru [38]. The possible reason could be urban children are at greater risk of obesity due to several factors specific to city environments. These include increased access to energy-dense foods like fast food and sugary drinks, a growing shift towards unhealthy western diets, and more sedentary lifestyles, which are worsened by limited outdoor spaces and more screen time [3]. These environmental and behavioral factors contribute to higher obesity rates compared to rural areas. Healthcare professionals in urban settings play a key role in detecting and preventing childhood obesity early. They must advocate for and implement interventions that promote healthy eating, encourage physical activity, and address socio-economic barriers to health in order to reduce obesity rates among urban children [3].

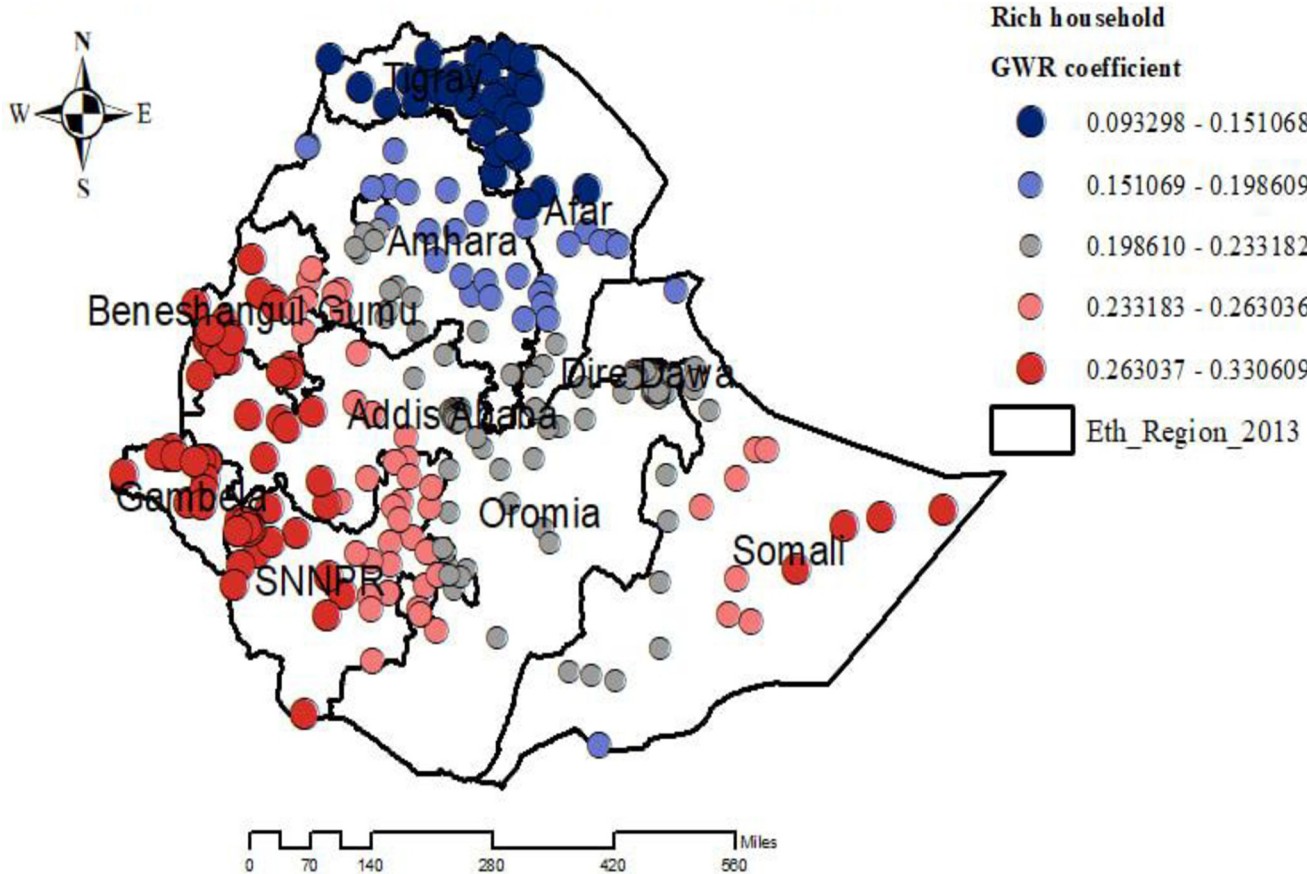

**Fig 8. Geographically weighted regression coefficients of rich households to predict the hotspots of overweight/obesity among under-five children in Ethiopia.** EMDHS 2019.

In addition, a strong negative correlation was found between the proportion of female children and overweight/obesity among children under five in the regions of Tigray, Amhara, Northwest Afar, and Beneshangul Gumuz. Consequently, it was discovered that male children had higher odds of being overweight or obese than female children when examining the relationship between a child's sex and overweight or obesity. This finding was consistent with previous studies done in Ethiopia [3], East Africa [19], Ghana [39], and Cameron [28]. The gender differences in overweight and obesity among children in these regions are likely due to a complex mix of genetic, environmental, metabolic, nutritional, and behavioral factors. Genetic predispositions and metabolic differences may affect weight gain, while environmental and nutritional factor like access to healthy food play significant roles. Additionally, cultural practices and behavioral patterns further contribute to these disparities, highlighting the need for a multifaceted approach to address the issue effectively [19, 40, 41].

## Implication of the study

The study identifies specific regions in Ethiopia, such as southern Amhara, northwest Somalia, Harari border, central Addis Ababa, eastern SNNPR, and northwest Oromia as significant hotspots for childhood overweight/obesity. Practical implications include targeted interventions in these hotspots, such as allocating resources for recreational facilities and physical education programs to promote active lifestyles among children. Health education campaigns via mass

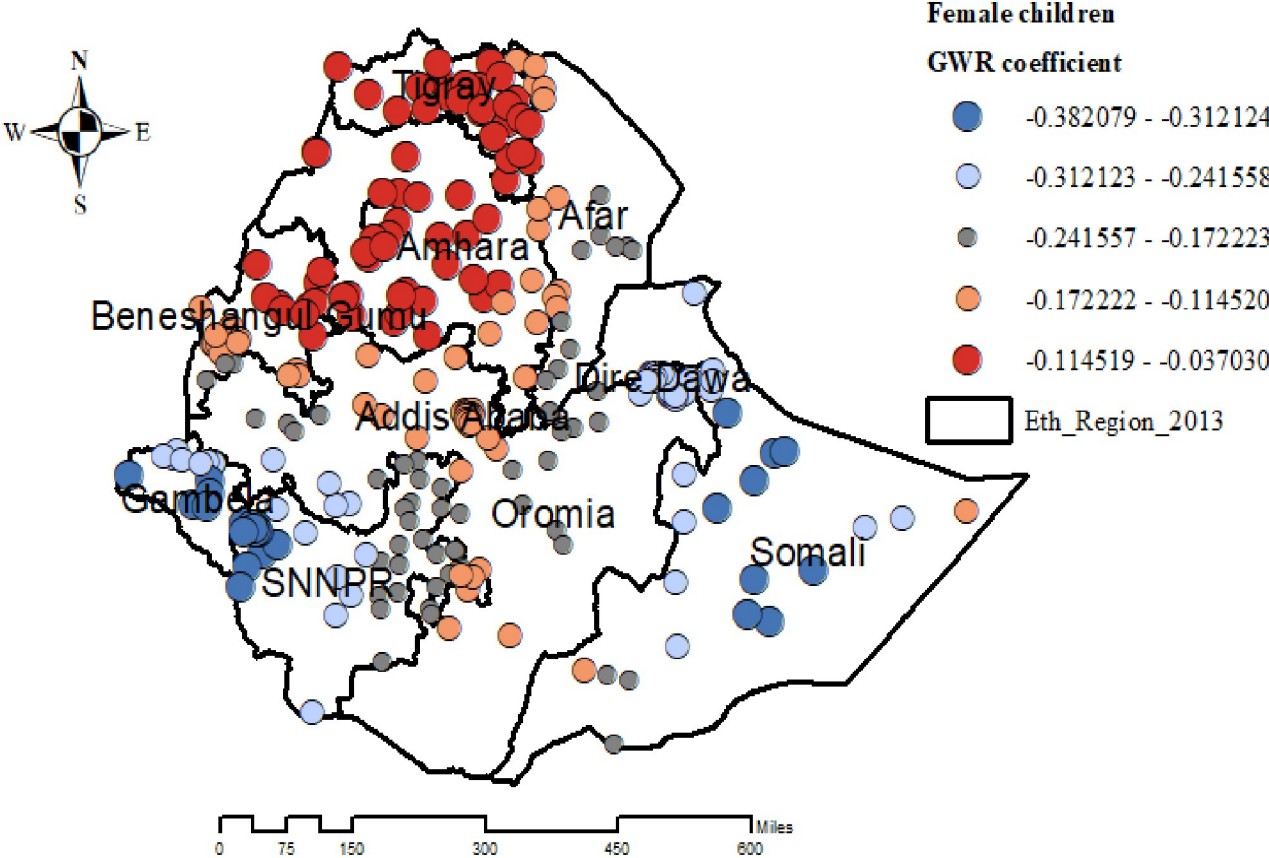

**Fig 9. Geographically weighted regression coefficients of child sex to predict the hotspots of overweight/obesity among under-five children in Ethiopia.** EMDHS 2019.

media are crucial for encouraging healthier behaviors like increased physical activity and improved dietary habits nationwide, tailored to local cultural contexts for maximum impact.

Further recommendations emphasize research to understand regional dietary patterns and nutritional needs among children, while training healthcare providers in hotspot regions to effectively manage and intervene early in childhood overweight/obesity cases. Community engagement is vital, involving local leaders and parents in promoting healthy environments and behaviors. Policymakers should collaborate to enact and enforce policies supporting children's health, including regulations on food marketing and ensuring access to nutritious foods in schools and communities. These measures collectively aim to mitigate childhood overweight/obesity disparities and enhance overall health outcomes across Ethiopia, providing a comprehensive framework for policymakers to address public health challenges effectively.

### Strengths and limitations of the study

The strength of this study was using a large and nationally representative EMDHS dataset, which was weighted and resulted in suitable statistical power that enables us to generalize these findings at the national level. Furthermore, the spatial and SaTScan-based cluster analyses were conducted for the detection of statistically significant high-risk clusters/hotspots of children overweight/obesity.

As a limitation, the study did not establish a causal relationship between the dependent and independent variables due to the cross-sectional nature of the demographic and health survey

data. Additionally, survey responses may be subject to recall bias, which could affect the accuracy of the reported information. This study also focused on specific predictors contributing to childhood overweight and obesity and it did not account for missing variables from the Ethiopian Demographic and Health Survey (EMDHS), which could have provided a more comprehensive understanding of other influencing predictors.

## Conclusions

Overweight/obesity among under-five children had spatial variations across the Ethiopian regions. Statistical analysis revealed that the southern portion of Amhara, northwest Somalia, the border of Harari, central Addis Ababa, the eastern portion of SNNPR, and the northwest portion of Oromia region were significant-high hotspots for child overweight/obesity. The GWR result revealed that in various regions of Ethiopia, the sex of the children, urban residency, household wealth index, and cesarean section delivery of children were all spatially significant determinants.

To effectively combat childhood obesity in Ethiopia, public health strategies must be tailored to the identified regional hotspots, such as southern Amhara and northwest Oromia, where higher obesity rates are observed. Policymakers should prioritize community-based initiatives that promote healthier lifestyles through nutritional counseling and awareness campaigns, especially targeting wealthier households. The Ministry of Health and the Ethiopian Public Health Institute (EPHI) should also invest in developing recreational facilities and physical education programs, particularly in urban areas where obesity is more prevalent. Public and private mass media can play a critical role in promoting active lifestyles and reducing sedentary behavior among children. Additionally, maternal and child health programs should focus on safe delivery practices and breastfeeding, as cesarean sections are linked to higher obesity rates.

Future research should include adjustments for confounders such as parental BMI to refine the observed associations. This approach will provide a more accurate understanding of the direct effects of urban residence and cesarean delivery on the outcome. And should be consider dietary habits, breastfeeding practices, maternal nutritional status, and physical activity levels of predictors for the representative of the finding. Moreover, longitudinal research is also essential to track trends and assess interventions, providing data on children's dietary patterns and food availability to inform policy refinement and ensure sustainable, region-specific solutions.

## Supporting information

**S1 Dataset.**
(XLSX)

## Acknowledgments

We would like to greatly acknowledge Measure DHS for providing the dataset for the study.

## Author Contributions

**Conceptualization:** Agmasie Damtew Walle, Shimels Derso Kebede, Jibril Bashir Adem, Ermias Bekele Enyew, Habtamu Alganeh Guadie, Teshome Bekana, Habtamu Setegn Ngusie, Sisay Maru Wubante, Sisay Yitayih Kassie, Addisalem Workie Demsash, Wabi Temesgen Atinafu, Tigist Andargie Ferede.

**Data curation:** Agmasie Damtew Walle, Shimels Derso Kebede, Jibril Bashir Adem, Ermias Bekele Enyew, Habtamu Alganeh Guadie, Teshome Bekana, Habtamu Setegn Ngusie, Sisay Maru Wubante, Sisay Yitayih Kassie, Addisalem Workie Demsash, Wabi Temesgen Atinafu, Tigist Andargie Ferede.

**Formal analysis:** Agmasie Damtew Walle, Shimels Derso Kebede, Jibril Bashir Adem, Teshome Bekana, Sisay Maru Wubante, Addisalem Workie Demsash, Wabi Temesgen Atinafu, Tigist Andargie Ferede.

**Funding acquisition:** Teshome Bekana, Habtamu Setegn Ngusie, Tigist Andargie Ferede.

**Investigation:** Agmasie Damtew Walle, Shimels Derso Kebede, Wabi Temesgen Atinafu, Tigist Andargie Ferede.

**Methodology:** Agmasie Damtew Walle, Shimels Derso Kebede, Jibril Bashir Adem, Ermias Bekele Enyew.

**Project administration:** Teshome Bekana, Tigist Andargie Ferede.

**Resources:** Agmasie Damtew Walle, Teshome Bekana.

**Software:** Agmasie Damtew Walle, Habtamu Setegn Ngusie.

**Supervision:** Agmasie Damtew Walle, Habtamu Alganeh Guadie, Teshome Bekana, Habtamu Setegn Ngusie.

**Validation:** Agmasie Damtew Walle, Habtamu Alganeh Guadie, Teshome Bekana, Habtamu Setegn Ngusie, Tigist Andargie Ferede.

**Visualization:** Agmasie Damtew Walle, Habtamu Alganeh Guadie, Teshome Bekana, Wabi Temesgen Atinafu, Tigist Andargie Ferede.

**Writing – original draft:** Agmasie Damtew Walle, Shimels Derso Kebede, Jibril Bashir Adem, Ermias Bekele Enyew, Habtamu Alganeh Guadie, Teshome Bekana, Habtamu Setegn Ngusie, Sisay Maru Wubante, Sisay Yitayih Kassie, Addisalem Workie Demsash, Wabi Temesgen Atinafu, Tigist Andargie Ferede.

**Writing – review & editing:** Agmasie Damtew Walle, Shimels Derso Kebede, Jibril Bashir Adem, Ermias Bekele Enyew, Habtamu Alganeh Guadie, Teshome Bekana, Habtamu Setegn Ngusie, Sisay Maru Wubante, Sisay Yitayih Kassie, Addisalem Workie Demsash, Wabi Temesgen Atinafu, Tigist Andargie Ferede.

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
