## [Decision Letter · Decision Letter 0]

26 Apr 2024

PONE-D-23-34960Spatial variations and associated factors of overweight/obesity among under-five children in Ethiopia: application of geographically weighted regression analysisPLOS ONE

Dear Dr. Walle,

Thank you for submitting your manuscript to PLOS ONE. After careful consideration, we feel that it has merit but does not fully meet PLOS ONE’s publication criteria as it currently stands. Therefore, we invite you to submit a revised version of the manuscript that addresses the points raised during the review process.

We look forward to receiving your revised manuscript.

Kind regards,

Clement Ameh Yaro, Ph.D

Academic Editor

PLOS ONE

Journal Requirements:

3. We note that Figures 3-9 in your submission contain map/satellite images which may be copyrighted. All PLOS content is published under the Creative Commons Attribution License (CC BY 4.0), which means that the manuscript, images, and Supporting Information files will be freely available online, and any third party is permitted to access, download, copy, distribute, and use these materials in any way, even commercially, with proper attribution. For these reasons, we cannot publish previously copyrighted maps or satellite images created using proprietary data, such as Google software (Google Maps, Street View, and Earth). For more information, see our copyright guidelines: http://journals.plos.org/plosone/s/licenses-and-copyright.

a. You may seek permission from the original copyright holder of Figures 3-9 to publish the content specifically under the CC BY 4.0 license.  

Reviewers' comments:

Reviewer's Responses to Questions

**Comments to the Author**

1. Is the manuscript technically sound, and do the data support the conclusions?

Reviewer #1: Partly

Reviewer #2: Partly

2. Has the statistical analysis been performed appropriately and rigorously? 

Reviewer #1: Yes

Reviewer #2: No

3. Have the authors made all data underlying the findings in their manuscript fully available?

Reviewer #1: Yes

Reviewer #2: Yes

4. Is the manuscript presented in an intelligible fashion and written in standard English?

Reviewer #1: Yes

Reviewer #2: No

5. Review Comments to the Author

Reviewer #1: Overweight review

About the novelty of the title

I can see that the title ha has been published through various ways. Hence, the authors should have a justification regarding it (few of them are listed below based on EDHS 2019,)

1. Spatial variation and determinants of underweight among children under 5 y of age in Ethiopia: A multilevel and spatial analysis based on data from the 2019 Ethiopian Demographic and Health Survey

2. Spatial distribution and associated factors of severe malnutrition among under-five children in Ethiopia: further analysis of 2019 mini-EDHS

3. Prevalence of overweight/obesity and associated factors among under-five children in Ethiopia: A multilevel analysis of nationally representative sample

Title

Suggestions for Improvement:

Consider including the specific timeframe of the study. For example, if the data collection period is mentioned, it can provide additional context and make the title more precise.

To enhance clarity, consider specifying the exact type of geographically weighted regression analysis utilized in the study. This will provide more detail on the specific approach used and could be of interest to readers familiar with different variations of geographically weighted regression.

Revised Example:

"Spatial variations and associated factors of overweight/obesity among under-five children in Ethiopia: a geographically weighted regression analysis of 2019 DHS data"

Abbreviation/acronyms utilization

Starting from your abstract the authors should re-arrange the utilization of abbreviations and acronyms accordingly

Introduction

The introduction section provides an overview of the global prevalence of childhood obesity/overweight and highlights the increasing burden in low-income countries, specifically focusing on Ethiopia. Here are some comments on the introduction:

Outdated references: The introduction section relies on references that are outdated, as indicated by the reference numbers (e.g., reference numbers 1, 2, 3, etc.). It is important to ensure that the references used are current and reflect the latest available data and research on the topic. Some of the reference also incorrectly cited. The authors should find the primary source and let the authors acknowledge them

Lack of global to local context: The introduction does not provide a clear transition from the global context of childhood obesity/overweight to the specific local context of Ethiopia. It would be beneficial to link the global prevalence statistics to the situation in Ethiopia and highlight the unique factors contributing to childhood obesity/overweight in the country.

Limited attention to childhood obesity in Ethiopia: The introduction suggests that Ethiopia does not consider childhood obesity as an emergent public health concern and mentions limited attention given to this issue. However, they did not put the references for their investigation or justification, and it would be helpful to provide more context or evidence to support this claim and explain the potential reasons for the lack of attention.

Need for spatial analysis: The introduction mentions that previous research in Ethiopia did not consider the spatial distribution of overweight/obesity and spatial regression analysis. It correctly highlights the importance of spatial analysis for policymakers to allocate resources effectively. However, it would be beneficial to provide more context on why spatial analysis is specifically relevant in the Ethiopian context and how it can contribute to addressing the issue.

Research gap: The introduction identifies a research gap by stating that few studies have explored community-level factors influencing overweight/obesity in Ethiopia. This highlights the need for the current study and its contribution to filling this gap. However, it would be helpful to provide more context on the specific aspects of community-level factors that have been underexplored in previous research.

Importance of the study: The introduction briefly mentions that the study aims to determine the spatial variations of overweight/obesity and its associated factors among under-five children in Ethiopia. It also emphasizes the potential implications of the study for policymakers, health planners, researchers, and health professionals. However, it would be beneficial to provide more specific details on how the findings of this study can inform policy and practice in Ethiopia.

Methods and materials

Study design, period, and area

What is EMDHS

Please clearly define the study setting, and related concepts or feeding practices and sources in the country. It will help readers

Try to put regionals states based on their order, and indicate there also another additional emerging region

Source and study population

The authors declared that their population was from 0-59 months. The question is

Body Mass Index (BMI) is generally not measured for newborns. BMI is a measure of body fat based on height and weight, typically used in older children, adolescents, and adults. For newborns, different measures are used to assess growth and development, such as weight-for-age, length-for-age, and weight-for-length percentiles. These measures help evaluate whether a newborn's growth is within the expected range for their age and sex.

BMI calculations require both height and weight measurements, which are not practical or accurate to obtain in newborn infants. Additionally, BMI is not considered an appropriate indicator of body fat for infants due to their rapidly changing body composition and growth patterns.

What about those children whose day of birth is missing or unknown

How did you manage other missing values. The authors should put the detail regarding their management

Data collection tool and procedures

In this section the author mentioned about other than the given topic

No sample size determination and sampling methods section

The authors need to create new session about sample size determination and sampling methods

Reference 14, 19…were not the appropriate references for the Data collection tool and procedures and outcome variable. Kindly use the correct one

Data management and statistical analysis

What does it mean “Using the STATA drop command in conjunction with a logical or conditional expression, we remove missing values from our analysis”.

Autocorrelation

Why the authors applied “The outcome variable has a Bernoulli distribution”. Have you appreciated that the continuous nature was also used in your model? Why did not use the poison one

Results

Do not use the new sentence with “of” and the like

How many unweighted participants did you included

Have the authors found primary clusters only in the SaTscan analysis

“Spatial scan analysis” what do you mean by “Spatial scan analysis” did you mean “Spatial SaTScan analysis”

Discussion

The provided discussion contains valuable information about the geographical variations in overweight/obesity and related factors among Ethiopian children under the age of five. However, there are a few areas that could be improved:

Consistency in referencing: The discussion refers to various studies to compare the prevalence of overweight/obesity in Ethiopian children, but the references are not consistently cited using proper citation format. It would be helpful to ensure that the references are cited correctly and consistently throughout the discussion. There are also miss citations. The author should have the primary source for their citation

Clarity in reporting findings: While the discussion provides some numerical findings regarding the prevalence of overweight/obesity in different regions, it would be beneficial to present the findings in a more organized and structured manner. Consider using tables or concise bullet-point format to clearly present the prevalence rates and comparisons between regions and previous studies.

Explanation of findings: The discussion mentions various factors that may explain the observed variations in overweight/obesity, such as delivery via cesarean section, socioeconomic status, and urban residence. It would be helpful to provide more detailed explanations for each factor, including the underlying mechanisms and potential implications for interventions or preventive measures.

Expansion of limitations and implications: The discussion briefly mentions methodological differences, cultural feeding practices, and nutrition assessment methods as possible explanations for the observed discrepancies in overweight/obesity prevalence. It would be beneficial to expand on these limitations and discuss their implications for future research or interventions in more detail. Furthermore, it was not clear how and why authors excluded missing values

Consistency in language and terminology: Some sentences in the discussion use different terminology or language styles. It is important to maintain consistency in terminology and language throughout the discussion to ensure clarity and coherence

Write clearly the implication of your study findings to the policymakers

Reviewer #2: Abstract:

1. Result : please support by statistics the following result “The spatial distribution of overweight/obesity among under-five children in Ethiopia was clustered”

2. Please support by statistics the following finding in the result “In the geographically weighted regression analysis, urban residence, cesarean section, rich households, and male children were statistically significant factors.

3. conclusion: please put yours recommendation based on the result , means against the identified factors

4. what is the contribution of 12 authors in secondary analysis

Background

1. please put citation for the following statement “The majority of overweight and obese children live in developing countries, where the pace of increase is more than 30% greater than in industrialized countries.

2. The following statement is not clear please rewrite “Globally, the prevalence of children under five who are obese was 7% in 2012, and by 2025, it was expected to drop to less than 11% (5)” can we say 7% to 11 % is drop???.

3. Please rephrase the following statements “From 5.4 million in 1990 to 10.3 million in 2014, the number of overweight and obese children in Africa has almost doubled (2, 6)”

4. Rephrase the following statement as “6.8% of children between the ages of 0 and 59 months are overweight or obese, according to 26 Demographic and Health Surveys that have been carried out in the SSA since 2010 (7) “.you can rephrase as : according to the Demographic and Health Surveys conducted in SSA, about 6.8% of children between the ages of 0 and 59 months were overweight or obese in 2010.

Generally, the introduction is not well written, incoherence, and many of the statement were written without citation, my recommendation for the authors, please remove unnecessary details and put your evidence or citation for some of statements that reported without citation.

Methods:

1. When do you score "1," else marked as "0" for the outcome variable “Overweight/obesity “

2. The following statement is not clear “STATA was used to tabulate the weighted proportions of outcome variables and prospective predictor variables, which were then exported to Excel and imported into ArcGIS 10.6 for additional analysis”, please remove

3. How the authors handle the two levels hierarchy nature of MEDHS data (multi-stage stratified cluster). That the Level one units were individual children in households and level two units were enumeration areas (community level). That Level one children were nested in the households, then the households were nested at the next higher level of enumeration areas ( community level). With is reality, Is Geographic Weighted Regression model is the appropriate model to identify the individual level and community-level factors as you claim in the bottom of your introduction???. I firmly recommend the authors to run multilevel mixed effect logistic regression rather Geographic Weighted Regression model to identify the individual level and community-level factors as you claim to identify factors at respective level.

4. How you weight the variables

Results

5. Please, specify the individual level and community-level factors separately in result

6. Discussion :generally the discussion need rewriting and grammar correction

7. Please write the clinical and public health importance of this study ,just at end of discussion

6. PLOS authors have the option to publish the peer review history of their article (what does this mean?). If published, this will include your full peer review and any attached files.

Reviewer #1: No

Reviewer #2: No

---

## [Author Response · Author response to Decision Letter 0]

18 Jul 2024

Date: 08/07/2024

Dear editorial board member (s) of PLOS ONE journal

The authors have been recalled to revise the manuscript entitled “Spatial variations and associated factors of overweight/obesity among under-five children in Ethiopia: a geographically weighted regression analysis of 2019 DHS data” identified by unique manuscript number: PONE-D-23-34960, which was submitted for publication. So, we received the editor(s)’ and reviewer's revision comments for the improvement of the manuscript before its publication.

Thank you for receiving comments, suggestions, directions, and questions from the editor and reviewer. As we said usually, we are very happy in receiving constructive and valuable comments for making a better manuscript. Accordingly, we have considered all the comments, questions, directions, and suggestions and provided a point-by-point response letter. 

Finally, we have submitted all the required documents in their revised form. We hope that we have addressed all the suggestions, directions, and raised questions and if you believe that point(s) is not addressed, please let us know. 

Thank you very much to all editor (s), and reviewers

On the behalf of the authors

Yours sincerely,

Correspondence author

PLOS ONE academic editor

Point-by-point response letter 

Reviewer comments 1

About the novelty of the title 

I can see that the title ha has been published through various ways. Hence, the authors should have a justification regarding it (few of them are listed below based on EDHS 2019,)

1. Spatial variation and determinants of underweight among children under 5 y of age in Ethiopia: A multilevel and spatial analysis based on data from the 2019 Ethiopian Demographic and Health Survey

2. Spatial distribution and associated factors of severe malnutrition among under-five children in Ethiopia: further analysis of 2019 mini-EDHS

3. Prevalence of overweight/obesity and associated factors among under-five children in Ethiopia: A multilevel analysis of nationally representative sample

Author reply: Thank you for your essential comment. Even though the titles was done by multilevel analysis, this study focused on Geographically Weighted Regression (GWR) analysis that allows for a nuanced understanding of the spatial variations in factors contributing to overweight among children under five, revealing how relationships between overweight prevalence and predictors like socioeconomic status, dietary patterns, physical activity, and environmental factors differ across geographic locations. This localized insight identifies specific hotspots and cold spots, enabling policymakers to tailor interventions to regional needs, thus optimizing resource allocation and intervention strategies. By providing detailed spatial patterns and enhancing understanding of underlying mechanisms, GWR supports the development of targeted, effective, and contextually appropriate health policies and programs.

Title 

Suggestions for Improvement:

Consider including the specific timeframe of the study. For example, if the data collection period is mentioned, it can provide additional context and make the title more precise.

To enhance clarity, consider specifying the exact type of geographically weighted regression analysis utilized in the study. This will provide more detail on the specific approach used and could be of interest to readers familiar with different variations of geographically weighted regression.

Revised Example:

"Spatial variations and associated factors of overweight/obesity among under-five children in Ethiopia: a geographically weighted regression analysis of 2019 DHS data"

Author reply: thank you for your fruitful comment and based on your comment we revised the title accordingly.

Abbreviation/acronyms utilization

Starting from your abstract the authors should re-arrange the utilization of abbreviations and acronyms accordingly 

Author reply: thank you for your suggestion and based on your comment we revised the the abbreviation and acronym accordingly.

Introduction 

The introduction section provides an overview of the global prevalence of childhood obesity/overweight and highlights the increasing burden in low-income countries, specifically focusing on Ethiopia. Here are some comments on the introduction:

Author reply: Thank you for your input. We presented the statement of the problem starting from a global perspective, then focusing on Africa, and finally narrowing down to our specific study context. We aimed to describe the prevalence, risk factors, and burden of overweight in a hierarchical manner. Thank you!

Outdated references: The introduction section relies on references that are outdated, as indicated by the reference numbers (e.g., reference numbers 1, 2, 3, etc.). It is important to ensure that the references used are current and reflect the latest available data and research on the topic. Some of the reference also incorrectly cited. The authors should find the primary source and let the authors acknowledge them 

Author reply: thank you for your suggestion and based on your comment we revised the reference by incorporating only the recent references thank you!

Lack of global to local context: The introduction does not provide a clear transition from the global context of childhood obesity/overweight to the specific local context of Ethiopia. It would be beneficial to link the global prevalence statistics to the situation in Ethiopia and highlight the unique factors contributing to childhood obesity/overweight in the country.

Author reply: thank you for your suggestion and based on your comment we revised the introduction section to include context from the global to the Ethiopian perspective, highlighting previous studies.

Limited attention to childhood obesity in Ethiopia: The introduction suggests that Ethiopia does not consider childhood obesity as an emergent public health concern and mentions limited attention given to this issue. However, they did not put the references for their investigation or justification, and it would be helpful to provide more context or evidence to support this claim and explain the potential reasons for the lack of attention.

Author reply: thank you for your suggestion and based on your comment we revised the introduction section to include childhood obesity as an emergent public health concern and mentions limited attention given to this issue with updated reference.

Need for spatial analysis: The introduction mentions that previous research in Ethiopia did not consider the spatial distribution of overweight/obesity and spatial regression analysis. It correctly highlights the importance of spatial analysis for policymakers to allocate resources effectively. However, it would be beneficial to provide more context on why spatial analysis is specifically relevant in the Ethiopian context and how it can contribute to addressing the issue.

Author reply: thank you for your fruit full comment and based on your comment we revised the introduction section by incorporating the need of spatial analysis for the study. Please have a look. Thank you!

Research gap: The introduction identifies a research gap by stating that few studies have explored community-level factors influencing overweight/obesity in Ethiopia. This highlights the need for the current study and its contribution to filling this gap. However, it would be helpful to provide more context on the specific aspects of community-level factors that have been underexplored in previous research.

Author reply: We appreciate your comment, which has guided the revision of our introduction. In response to identified gaps in existing literature, our study utilizes secondary data from the Demographic and Health Surveys (DHS), incorporating a comprehensive array of relevant variables. Our study emphasizes the underexplored community-level factors which included in the data were consider that influences on overweight and obesity in Ethiopia. By employing Geographically Weighted Regression (GWR), we analyze spatial variations in these factors across different regions, aiming to provide nuanced insights that inform targeted public health interventions tailored to local contexts.

Importance of the study: The introduction briefly mentions that the study aims to determine the spatial variations of overweight/obesity and its associated factors among under-five children in Ethiopia. It also emphasizes the potential implications of the study for policymakers, health planners, researchers, and health professionals. However, it would be beneficial to provide more specific details on how the findings of this study can inform policy and practice in Ethiopia.

Author reply: Thank you for your best and fruit full comment based on this we revised as, the study aims to investigate spatial variations of overweight/obesity and associated factors among under-five children in Ethiopia, utilizing Geographically Weighted Regression (GWR) with Demographic and Health Survey (DHS) data. The findings hold significant implications for policymakers, health planners, researchers, and health professionals. Policymakers can use the results to develop targeted policies and interventions to reduce childhood overweight/obesity rates across different regions. Health planners will benefit from insights to strategically allocate resources, focusing on areas with higher prevalence and specific risk factors identified by the study. Researchers can build upon these findings to explore causal relationships and refine intervention strategies, while health professionals can implement evidence-based approaches tailored to local contexts. Ultimately, these actions can collectively contribute to improving public health outcomes for under-five children in Ethiopia by addressing both prevention and management of overweight/obesity.

Methods and materials 

Study design, period, and area

What is EMDHS

Author reply: thank you for your concern and for more understanding of the abbreviation we were incorporate in the abbreviation section as Ethiopian Mini Demographic Health Survey.

Please clearly define the study setting, and related concepts or feeding practices and sources in the country. It will help readers Try to put regionals states based on their order, and indicate there also another additional emerging region 

Author reply Thank you for your suggestion and based on your comment we were describe the study setting clearly in the method section, thank you

Source and study population

The authors declared that their population was from 0-59 months. The question is

Body Mass Index (BMI) is generally not measured for newborns. BMI is a measure of body fat based on height and weight, typically used in older children, adolescents, and adults. For newborns, different measures are used to assess growth and development, such as weight-for-age, length-for-age, and weight-for-length percentiles. These measures help evaluate whether a newborn's growth is within the expected range for their age and sex.

BMI calculations require both height and weight measurements, which are not practical or accurate to obtain in newborn infants. Additionally, BMI is not considered an appropriate indicator of body fat for infants due to their rapidly changing body composition and growth patterns.

Author reply: Thank you for your concern and in our study the outcome was measured using the World Health Organization (WHO) standards for child growth monitoring. Specifically, they used the weight-for-height z-score, which is a reliable method for assessing growth and nutritional status in children. According to WHO guidelines, a child is classified as overweight or obese if their weight-for-height z-score exceeds +2.0 standard deviations (SD) from the mean. This approach allows for accurate monitoring of growth and development, ensuring that children who are at risk of overweight or obesity are correctly identified based on globally recognized standards. This method accounts for the rapidly changing body composition and growth patterns in infants and young children, making it a suitable alternative to BMI for this age group.

What about those children whose day of birth is missing or unknown.

How did you manage other missing values? The authors should put the detail regarding their management 

Author reply: Thank you for your concern and we were handling missing data, imputation techniques are employed to estimate or replace missing values based on existing data. Methods like mean, mode, or median imputation replace missing values with summary statistics of observed data, assuming similarity among values. More advanced techniques include regression-based imputation, where missing values are predicted using relationships with other variables. 

 Data collection tool and procedures

In this section the author mentioned about other than the given topic 

No sample size determination and sampling methods section 

The authors need to create new session about sample size determination and sampling methods 

Reference 14, 19…were not the appropriate references for the Data collection tool and procedures and outcome variable. Kindly use the correct one 

Author reply: thank you for your suggestion and we revised it accordingly. Thank you!

Data management and statistical analysis

 What does it mean “Using the STATA drop command in conjunction with a logical or conditional expression, we remove missing values from our analysis”.

Author reply: thank you very much for your input to improve the manuscript. Accordingly, we revised it clearly.

Autocorrelation 

Why the authors applied “The outcome variable has a Bernoulli distribution”. Have you appreciated that the continuous nature was also used in your model? Why did not use the poison one

Author reply: thank you for your best concern. In our study, the decision to model the outcome variable with a Bernoulli distribution instead of a continuous distribution like Poisson was guided by the nature of the data and statistical considerations. The outcome variable in our analysis is binary, representing a dichotomous response (yes and no). A Bernoulli distribution is well-suited for such binary outcomes as it directly models the probability of a single binary event occurring. While the outcome variable is binary, our modeling approach incorporates continuous predictors or covariates to capture relationships with the probability of the binary outcome. This enables us to examine how these predictors influence the likelihood of the outcome, enhancing both the interpretability and predictive accuracy of our model. Comparisons during model selection supported the use of the Bernoulli distribution over alternatives like Poisson, aligning with the characteristics of our data and facilitating clearer interpretation of model results in the context of binary outcomes. Thank you!

Results 

Do not use the new sentence with “of” and the like 

Author reply: ok thank you!

How many unweighted participants did you included 

Author reply: thank you for your concern and before weighting the participant’s number we have got 3,602. It was below the weighted participants.

Have the authors found primary clusters only in the SaTscan analysis

“Spatial scan analysis” what do you mean by “Spatial scan analysis” did you mean “Spatial SaTScan analysis”

Author reply: yes, we want to write as Spatial SaTScan analysis, so we were revised it clearly, thank you!

Discussion 

The provided discussion contains valuable information about the geographical variations in overweight/obesity and related factors among Ethiopian children under the age of five. However, there are a few areas that could be improved:

Consistency in referencing: The discussion refers to various studies to compare the prevalence of overweight/obesity in Ethiopian children, but the references are not consistently cited using proper citation format. It would be helpful to ensure that the references are cited correctly and consistently throughout the discussion. There are also miss citations. The author should have the primary source for their citation 

Clarity in reporting findings: While the discussion provides some numerical findings regarding the prevalence of overweight/obesity in different regions, it would be beneficia

---

## [Decision Letter · Decision Letter 1]

9 Sep 2024

PONE-D-23-34960R1Spatial variations and associated factors of overweight/obesity among under-five children in Ethiopia: a geographically weighted regression analysis of 2019 DHS dataPLOS ONE

Dear Dr. Walle,

Thank you for submitting your manuscript to PLOS ONE. After careful consideration, we feel that it has merit but does not fully meet PLOS ONE’s publication criteria as it currently stands. Therefore, we invite you to submit a revised version of the manuscript that addresses the points raised during the review process.

We look forward to receiving your revised manuscript.

Kind regards,

Clement Ameh Yaro, Ph.D

Academic Editor

PLOS ONE

Journal Requirements:

Reviewers' comments:

Reviewer's Responses to Questions

**Comments to the Author**

1. If the authors have adequately addressed your comments raised in a previous round of review and you feel that this manuscript is now acceptable for publication, you may indicate that here to bypass the “Comments to the Author” section, enter your conflict of interest statement in the “Confidential to Editor” section, and submit your "Accept" recommendation.

Reviewer #3: All comments have been addressed

Reviewer #4: (No Response)

2. Is the manuscript technically sound, and do the data support the conclusions?

Reviewer #3: Yes

Reviewer #4: Yes

3. Has the statistical analysis been performed appropriately and rigorously? 

Reviewer #3: No

Reviewer #4: (No Response)

4. Have the authors made all data underlying the findings in their manuscript fully available?

Reviewer #3: Yes

Reviewer #4: Yes

5. Is the manuscript presented in an intelligible fashion and written in standard English?

Reviewer #3: No

Reviewer #4: (No Response)

6. Review Comments to the Author

Reviewer #3: Title: Please write DHS's full name in the title; it is recommended that abbreviations be avoided.

Introduction: Kindly add the clear SMART objectives of the study at the end of the introduction/ background.

Methods: The study omitted several potentially influential factors, such as dietary habits, breastfeeding practices, maternal nutritional status, and physical activity levels. Including these variables would provide a more comprehensive analysis of the determinants of childhood obesity.

The paper could benefit from more detailed descriptions of the statistical methods, particularly the GWR analysis. Readers unfamiliar with these techniques might find that the current explanation needs to be revised. Including a brief introduction to how these methods work and why they were chosen would enhance understanding.

Results and Analysis: While the study identifies significant factors like urban residence and cesarean delivery, further adjustments for potential confounders (e.g., parental BMI, socio-economic status) could refine these associations

Conclusion: The conclusions could benefit from including more actionable steps or strategies for stakeholders. Suggestions for immediate actions based on the findings would make the study more practical and applicable.

Highlighting specific areas for future research, such as longitudinal studies to track changes over time or interventions tailored to specific regions, would provide a clear direction for advancing the understanding of childhood obesity.

Reviewer #4: Editor Comments:

The manuscript presents an interesting study on the effects of inulin on Spatial variations and associated factors of overweight/obesity among under-five children in Ethiopia: a geographically weighted regression analysis of 2019 DHS data. The manuscript is well written and comprehensive, and the methodology used seems robust as the sample is large and covers many locations in Ethiopia. However, there are some points that require revision to enhance the clarity and scientific accuracy of the manuscript.

•The introduction section provides a good overview of the global and regional context of childhood obesity. However, I suggest adding more recent data or references to give a more current picture of the situation. I would also recommend moving from the global context to the specific local context of Ethiopia to understand the factors contributing to obesity in Ethiopian children.

•In the results section, clarify the reporting of statistical analyses. For example, when discussing clustering patterns (Moran’s I and Z scores), consider providing more context on what these values indicate in practical terms.

•The section on “Factors influencing spatial variation in obesity/overweight” could benefit from more clearly separating individual-level and community-level factors, or at least providing a clearer explanation of why these factors are presented together.

•In the discussion section: I suggest providing a more explanation of the possible mechanisms underlying the associations of factors such as cesarean delivery, urban residence, and wealth index. These factors may provide a better explanation of childhood overweight/obesity.

•Finally, I suggest that rather than summarizing the main findings of the research in a general way in the conclusion, highlight how the study’s insights can be used to develop public health strategies to combat childhood obesity in Ethiopia, and provide more specific recommendations that policymakers can take based on the study’s findings.

•Ensure that all abbreviations are defined on first use and check for consistency throughout the manuscript.

Following the suggested comments, the manuscript will become more clear and impactful about the spatial patterns of childhood obesity in Ethiopia.

Best regards,

7. PLOS authors have the option to publish the peer review history of their article (what does this mean?). If published, this will include your full peer review and any attached files.

Reviewer #3: **Yes: **A Alyafei

Reviewer #4: **Yes: **Dr. Mai Albaik

---

## [Author Response · Author response to Decision Letter 1]

13 Sep 2024

Date: 10/09/2024 

Dear editorial board member (s) of PLOS ONE journal

The authors have been recalled to revise the manuscript entitled “Spatial variations and associated factors of overweight/obesity among under-five children in Ethiopia: a geographically weighted regression analysis of 2019 DHS data” identified by unique manuscript number: PONE-D-23-34960, which was submitted for publication. So, we received the editor(s)’ and reviewer's minor revision comments for the improvement of the manuscript before its publication.

Thank you for receiving comments, suggestions, directions, and questions from the editor and reviewer. As we said usually, we are very happy in receiving constructive and valuable comments for making a better manuscript. Accordingly, we have considered all the comments, questions, directions, and suggestions and provided a point-by-point response letter. 

Finally, we have submitted all the required documents in their revised form. We hope that we have addressed all the suggestions, directions, and raised questions and if you believe that point(s) is not addressed, please let us know. 

Thank you very much to all editor (s), and reviewers

On the behalf of the authors

Yours sincerely,

Correspondence author

PLOS ONE academic editor

Point-by-point response letter 

Editor comments 

The manuscript presents an interesting study on Spatial variations and associated factors of overweight/obesity among under-five children in Ethiopia: a geographically weighted regression analysis of 2019 DHS data. The manuscript is well written and comprehensive, and the methodology used seems robust as the sample is large and covers many locations in Ethiopia. However, there are some points that require revision to enhance the clarity and scientific accuracy of the manuscript.

• The introduction section provides a good overview of the global and regional context of childhood obesity. However, I suggest adding more recent data or references to give a more current picture of the situation. I would also recommend moving from the global context to the specific local context of Ethiopia to understand the factors contributing to obesity in Ethiopian children.

Author reply: Thank you for your fruitful comments and suggestion and based on your comment we revised the introduction section by incorporating global to Africa to Ethiopian context clearly please have a look. 

• In the results section, clarify the reporting of statistical analyses. For example, when discussing clustering patterns (Moran’s I and Z scores), consider providing more context on what these values indicate in practical terms.

Author reply: Thank you for your essential suggestion and we describe the indication of the Morans I and Z score in the result section line 346-349 as well as in the method section of spatial analysis heading, line 270-272, please have a look the highlighted colour in the tracked change of the manuscript. 

• The section on “Factors influencing spatial variation in obesity/overweight” could benefit from more clearly separating individual-level and community-level factors, or at least providing a clearer explanation of why these factors are presented together.

Author reply: Thank you for your concern and valuable feedback on the section Thank you for your feedback on the section “Factors Influencing Spatial Variation in Obesity/Overweight.” In this analysis, which utilizes Geographically Weighted Regression (GWR), we focused on identifying spatial variations in how different factors influence obesity/overweight rather than performing a traditional multilevel analysis. GWR allows us to examine how the impact of these factors varies across different geographic areas, providing insights into localized patterns and interactions. This approach is well-suited for capturing the spatial heterogeneity of factors and understanding how their effects differ regionally. To clarify, we were include a detailed explanation in the revised manuscript of how this method specifically addresses variations across different areas in introduction section clearly. Thank you!

• In the discussion section: I suggest providing a more explanation of the possible mechanisms underlying the associations of factors such as cesarean delivery, urban residence, and wealth index. These factors may provide a better explanation of childhood overweight/obesity.

Author reply: Thank you very much for your critical comment to improve the manuscript before publication and we revised the given comments in discussion section accordingly. Please have a look the discussion section of the manuscript. 

• Finally, I suggest that rather than summarizing the main findings of the research in a general way in the conclusion, highlight how the study’s insights can be used to develop public health strategies to combat childhood obesity in Ethiopia, and provide more specific recommendations that policymakers can take based on the study’s findings.

Author reply: Thank you very much for your suggestion to improve the manuscript before publication and we revised the conclusion section accordingly. Please have a look the conclusion section of the manuscript. 

• Ensure that all abbreviations are defined on first use and check for consistency throughout the manuscript.

Author reply: Thank you for your valuable suggestion regarding the definition and consistency of abbreviations. We have reviewed the manuscript to ensure that all abbreviations are defined upon first use and have checked for consistency throughout. As a result, all of them were consistent and defined clearly.

Reviewer 3 comments 

Reviewer #3: Title: Please write DHS's full name in the title; it is recommended that abbreviations be avoided.

Author reply: Thank you for your fruitful comments and suggestion and based on your comment we revised as “Spatial Variations and Predictors of Overweight/Obesity among Under-Five Children in Ethiopia: A Geographically Weighted Regression Analysis of the 2019 Ethiopian Mini Demographic and Health Survey “

Introduction: Kindly add the clear SMART objectives of the study at the end of the introduction/ background.

Author reply: Thank you for your fruitful comments and suggestion and based on your comment we revised the introduction section by incorporating SMART object at the end of the introduction section. Please have a look. 

Methods: The study omitted several potentially influential factors, such as dietary habits, breastfeeding practices, maternal nutritional status, and physical activity levels. Including these variables would provide a more comprehensive analysis of the determinants of childhood obesity.

Author reply: Thank you for your valuable comments to improve the manuscript before publication. Based on your feedback, we have revised the conclusion section by suggesting that future researchers include the factors you mentioned to enhance the representativeness of the findings. Kindly review the updated conclusion section.

The paper could benefit from more detailed descriptions of the statistical methods, particularly the GWR analysis. Readers unfamiliar with these techniques might find that the current explanation needs to be revised. Including a brief introduction to how these methods work and why they were chosen would enhance understanding.

Author reply: Thank you for your valuable feedback. Based on your suggestion, we have revised both the Introduction and Methods sections to provide a more detailed explanation of the Geographically Weighted Regression (GWR) analysis. In the introduction, we now include a brief overview of how GWR works and the rationale for selecting this method. In the methods section, we have expanded the description of the statistical process, explaining the steps taken in the GWR analysis, including how the model accounts for spatial heterogeneity and the reasoning behind the chosen parameters. These additions aim to enhance understanding for readers unfamiliar with GWR techniques.

Results and Analysis: While the study identifies significant factors like urban residence and cesarean delivery, further adjustments for potential confounders (e.g., parental BMI, socio-economic status) could refine these associations

Author reply: Thank you for your valuable comments to improve the manuscript before publication. Based on your feedback, we have revised the conclusion section by suggesting that future researchers include the factors you mentioned to enhance the representativeness of the findings. Kindly review the updated conclusion section.

Conclusion: The conclusions could benefit from including more actionable steps or strategies for stakeholders. Suggestions for immediate actions based on the findings would make the study more practical and applicable.

Highlighting specific areas for future research, such as longitudinal studies to track changes over time or interventions tailored to specific regions, would provide a clear direction for advancing the understanding of childhood obesity.

Author reply: Thank you very much for your suggestion to improve the manuscript before publication. We have revised the conclusion section to incorporate your feedback and have made the necessary adjustments for clarity. Please review the updated conclusion section of the manuscript.

With regards!

Agmasie Damtew Walle

Corresponding author

---

## [Decision Letter · Decision Letter 2]

1 Oct 2024

Spatial Variations and Predictors of Overweight/Obesity among Under-Five Children in Ethiopia: A Geographically Weighted Regression Analysis of the 2019 Ethiopian Mini Demographic and Health Survey

PONE-D-23-34960R2

Dear Dr. Walle,

We’re pleased to inform you that your manuscript has been judged scientifically suitable for publication and will be formally accepted for publication once it meets all outstanding technical requirements.

Kind regards,

Clement Ameh Yaro, Ph.D

Academic Editor

PLOS ONE

Additional Editor Comments (optional):

Reviewers' comments:

Reviewer's Responses to Questions

**Comments to the Author**

1. If the authors have adequately addressed your comments raised in a previous round of review and you feel that this manuscript is now acceptable for publication, you may indicate that here to bypass the “Comments to the Author” section, enter your conflict of interest statement in the “Confidential to Editor” section, and submit your "Accept" recommendation.

Reviewer #3: All comments have been addressed

Reviewer #4: All comments have been addressed

2. Is the manuscript technically sound, and do the data support the conclusions?

Reviewer #3: Yes

Reviewer #4: Yes

3. Has the statistical analysis been performed appropriately and rigorously? 

Reviewer #3: Yes

Reviewer #4: I Don't Know

4. Have the authors made all data underlying the findings in their manuscript fully available?

Reviewer #3: Yes

Reviewer #4: Yes

5. Is the manuscript presented in an intelligible fashion and written in standard English?

Reviewer #3: Yes

Reviewer #4: (No Response)

6. Review Comments to the Author

Reviewer #3: The study addresses the Innovative Use of Geospatial Techniques, which provide important insights into the spatial variations of childhood obesity and enable targeted intervention strategies. Furthermore, the study focuses on an emerging public health concern. Robust Analysis: The study’s use of a large, nationally representative dataset and advanced statistical techniques like the GWR model enhances the credibility of the findings and strengthens the implications for policy interventions.

Points for improvement in future studies to determine Causal Relationship: longitudinal data might help clarify the temporal relationship between identified predictors and childhood obesity. While the study provides excellent data on spatial variations, the intervention discussion could benefit from more specific policy recommendations. For example, what kind of localized physical education programs or nutritional campaigns would be most effective in the identified hot spot areas? Future studies could consider expanding on parental factors, such as parental BMI, education levels, and health behaviors, to provide a more nuanced understanding of risk factors.

Reviewer #4: The authors have carefully reviewed and made all the necessary corrections as requested. They have ensured that each of the suggested revisions has been addressed thoroughly, reflecting the feedback provided for the improvement of the manuscript.

7. PLOS authors have the option to publish the peer review history of their article (what does this mean?). If published, this will include your full peer review and any attached files.

Reviewer #3: **Yes: **Anees Alyafei

Reviewer #4: **Yes: **Dr. Mai Albaik

---

## [Editor Report · Acceptance letter]

4 Oct 2024

PONE-D-23-34960R2 

PLOS ONE

Dear Dr. Walle, 

I'm pleased to inform you that your manuscript has been deemed suitable for publication in PLOS ONE. Congratulations! Your manuscript is now being handed over to our production team.

Kind regards, 

on behalf of

Dr. Clement Ameh Yaro 

Academic Editor

PLOS ONE